# Comparative influenza protein interactomes identify the role of plakophilin 2 in virus restriction

Lingyan Wang[1,*], Bishi Fu[2,*], Wenjun Li[3], Girish Patil[1], Lin Liu[1], Martin E. Dorf[2] & Shitao Li[1]

Cellular protein interaction networks are integral to host defence and immune signalling pathways, which are often hijacked by viruses via protein interactions. However, the comparative virus–host protein interaction networks and how these networks control host immunity and viral infection remain to be elucidated. Here, we mapped protein interactomes between human host and several influenza A viruses (IAV). Comparative analyses of the interactomes identified common and unique interaction patterns regulating innate immunity and viral infection. Functional screening of the 'core' interactome consisting of common interactions identified five novel host factors regulating viral infection. Plakophilin 2 (PKP2), an influenza PB1-interacting protein, restricts IAV replication and competes with PB2 for PB1 binding. The binding competition leads to perturbation of the IAV polymerase complex, thereby limiting polymerase activity and subsequent viral replication. Taken together, comparative analyses of the influenza–host protein interactomes identified PKP2 as a natural inhibitor of IAV polymerase complex.

[1] Department of Physiological Sciences, Oklahoma State University, 264 McELroy Hall, Stillwater, Oklahoma 74078, USA. [2] Department of Microbiology & Immunobiology, Harvard Medical School, 77 Avenue Louis Pasteur, NRB830, Boston, Massachusetts 02115, USA. [3] School of Stomatology, Peking University, 22 Zhongguancun South Avenue, Beijing 100081, China. * These authors contributed equally to this work. Correspondence and requests for materials should be addressed to S.L. (email: shitao.li@okstate.edu).

nfluenza A virus (IAV) is a highly transmissible respiratory pathogen and presents a continued threat to global health, with considerable economic and social impact[1,2]. IAV is a member of the orthomyxoviridae family and possesses eight segments of a negative-sense single-stranded RNA genome. During IAV infection, host pattern recognition receptors, such as TLR7 and RIG-I, sense viral RNA and elicit interferon-mediated innate immunity to restrict IAV infection[3]. In addition, host intrinsic restriction factors impair IAV infection by interacting with viral proteins. For example, the E3 ligase TRIM32 ubiquitinates PB1, thereby leading to PB1 protein degradation and limiting viral infection[4]. By contrast, IAV proteins engage with the host cellular protein interaction network to hijack host molecular machinery to fulfil viral life cycle and perturb host defences to evade immune surveillance. Thus, the protein interactions between IAV and host contribute to the outcomes of viral pathogenesis.

IAV comprises a plethora of strains with different pathogenic profiles. Several recent proteomic studies identified a cohort of cellular factors that interact with IAV proteins[5–8]. However, the knowledge of common and strain-specific interactions is incomplete and how these interactions control host defence and viral infection remains to be fully elucidated. Systematic analysis of strain-specific IAV–host protein interactomes should reveal general and distinct mechanisms of regulating viral infection and host defence. Insights gained from these interactions will facilitate the design of future antiviral therapies.

Plakophilin 2 (PKP2) is the most prevalent plakophilin protein and essential for the formation of desmosomes and stabilisation of cell junctions[9]. Mutations in the human PKP2 gene have been linked to severe heart abnormalities leading to arrhythmogenic right ventricular cardiomyopathy, an inherited disorder of the cardiac muscle[10]. However, the role of PKP2 in viral infection is unknown.

In this study, we first used a proteomic approach to establish a comprehensive and dynamic interactome of 11 viral proteins of influenza A/Puerto Rico/8/1934 (H1N1) (PR8) in HEK293 cells. Analysis of the network revealed that M2, PB1, PB2 and NP are the major nodes connecting cellular factors with known and predicted roles in immunity and viral infection. Thus, we further mapped the protein interactomes of these 4 viral proteins plus NS1, the multifunctional viral protein, from two other H1N1 strains and one H5N1 strain. In addition, the protein inter-actomes of NS1 and NP from a H3N2 strain were mapped. Parallel comparisons of these interactomes revealed common and unique protein interaction patterns, suggesting general and distinct strategies of each viral strain. Gain- and loss-of-function studies of the common IAV interactors identified five novel host factors regulating viral infection. Our study further demonstrates that PKP2, a common PB1 interactor, inhibits the IAV polymerase activity, thereby limiting viral infection.

## Results
**Mapping IAV–host protein interactomes**. To uncover the comprehensive IAV–host protein interactions, we first performed affinity purification coupled with mass spectrometry (AP-MS) analysis of PR8 IAV protein complexes (Fig. 1a). Eleven C- and eight N-terminal FLAG-tagged viral genes were individually transfected into HEK293 cells to make stable cell lines (Supplementary Fig. 1a). The cell viability and virus growth of these stable cell lines were comparable (Supplementary Fig. 1b,c). Each stable HEK293 cell line was mock infected or infected with 1 multiplicity of infection (MOI) of PR8 IAV for 16 h. After infection, IAV protein complexes were affinity purified using anti-FLAG antibody and then analysed by mass spectrometry. Two biological repeats were obtained for each IAV

protein complex under IAV and mock infection conditions, respectively. To efficiently reduce false positives in AP-MS, we adopted the statistical method Significance Analysis of INTer-actome (SAINT)[11] in combination with the proteomic database derived from HEK293 cells in our laboratory (Supplementary Data 1–3)[12–14]. SAINT allows bench scientists to select bona fide interactions and remove non-specific interactions in an unbiased manner. Using a stringent statistical SAINT score ($\geq 0.89$), we identified 357 high-confidence candidate interacting proteins (HCIPs) in HEK293 cells, which forms a PR8 IAV–host interaction network of 529 protein interactions (Supplementary Fig. 2 and Supplementary Data 1; note that the data set for C-terminal FLAG-tagged PB1 in HEK293 without viral infection was reported previously[4]). Our network features 243 IAV-induced interactions, 88 IAV-suppressed interactions and 317 new interactions (Supplementary Data 4,5).

Systematic comparison of multiple influenza–host protein inter-actomes is essential for the identification of general mechanisms of regulation of viral infection and the discovery of common therapeutic targets. However, common and strain-specific virus–host protein interactions have not been systematically investigated. We therefore further mapped the protein interactomes of five IAV proteins (PB1, PB2, NP, NS1 and M2) from two additional H1N1 strains (WSN/33, A/WSN/1933 (H1N1); NY/2009, A/New York/1682/2009 (H1N1)) and one H5N1 strain (VN/2004, A/ /Viet Nam/1203/2004(H5N1)), plus the protein interactomes of NS1 and NP from a H3N2 strain (Aichi, A/Aichi/2/1968 (H3N2)) in HEK293 cells (Supplementary Data 6,7). NY/2009 is a pandemic strain, whereas VN/2004 is an emerging highly pathogenic strain[1,15]. VN/2004 viral protein complexes were only purified from uninfected stable cell lines. These viral proteins exhibited comparable expression levels in the stable cell lines (Supplementary Fig. 3a–e). The cell viability and virus growth of these stable cell lines were also comparable (Supplementary Fig. 4a,4b). These interactomes consist of 625 HCIPs, which is enriched in several pathways such as messenger RNA (mRNA) splicing, RNA transport and virus modulation (Supplementary Fig. 4c). Approximately 30% (304 of 1014) of the interactions were previously reported (Supplementary Data 8), including well-established interactors such as PIK3R2 (ref. 16), PPP6C[17], POLR2B[18] and IPO5 (ref. 19). Analysis of the interactomes revealed that the ratio of common HCIPs with each viral protein (of $\geq 2$ strains) varied from 26 to 60% (Fig. 1b). NP, PB1 and PB2 shared more than half of HCIPs among strains, whereas NS1 and M2 exhibited higher strain-specific interactions (Fig. 1b).

**Common and unique interaction patterns in the interactomes**. To comparatively analyse the interactomes, the relative protein abundance of HCIPs in each complex was determined by normalized spectral abundance factor[20]. We first analysed NS1 protein complexes from PR8, NY/2009, WSN/33, Aichi and VN/2004. Surprisingly, NS1 proteins displayed distinct protein interaction patterns (Fig. 1c). For example, the well-known NS1 interactors PIK3R2 and cleavage and polyadenylation specific factor 4 (CPSF4, also known as CPSF30) exhibited different compositions in NS1 complexes. PIK3R2 accounted for >45% of NS1 complexes of PR8, NY/2009 and VN/2004 but only a small fraction of NS1 complexes of WSN/33 ($\sim$15%) and Aichi ($\sim$5%) (Fig. 1c). By contrast, CPSF proteins and their complex components (FIP1L1 and WDR33) were abundant in NS1 complexes of WSN/33 and Aichi but absent in the PR8 NS1 complex (Fig. 1c and Supplementary Fig. 4d). These data are consistent with the absence of F103 and M106 in NS1 of PR8 influenza, which is required for stabilising the CPSF interaction[21]. The other pronounced interactors were PRPF39, prolyl 4-hydroxylases (P4HA1 and P4HB), and the tRNA-splicing

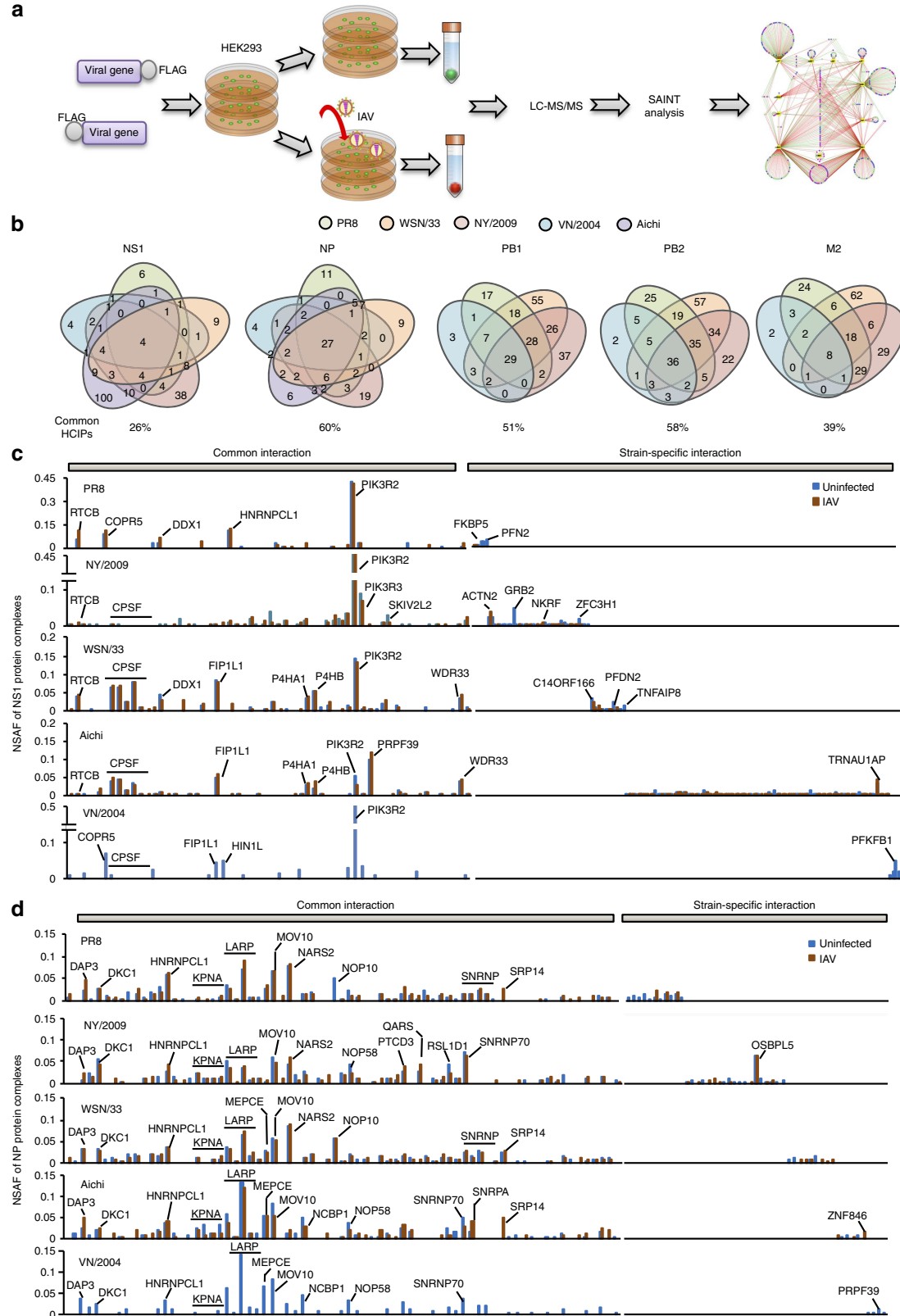

**Figure 1 | Comparative analysis of IAV–host interactomes. (a)** Pipeline of AP-MS for mapping IAV–host protein interactomes. **(b)** The Venn diagrams illustrate the number of common HCIPs (in ≥ 2 strains) in viral protein complexes from different IAV strains. The percentage of common HCIPs is indicated. **(c,d)** Comparison of the relative abundances of HCIPs of each NS1 **(c)** and NP **(d)** from five IAV strains. The outstanding HCIPs are labelled. NSAF stands for normalized spectral abundance factor.

ligase complex consisting of RTCB (also known as C22ORF28), DDX1 and C14ORF166 (ref. 22). In addition, each NS1 exhibited unique interactors in the strain-specific interaction, such as FKBP5, NKRF, TNFAIP8, ZFC3H1 and TRNAU1AP.

In contrast to NS1, the interactomes of NP, PB1, PB2 and M2 were more conserved among various IAV strains (Fig. 1d and Supplementary Fig. 5a–c). The NP interactomes featured common interactors involved in nuclear import (KPNA family proteins), transcription (LARP family proteins) and mRNA splicing (SNRNP proteins). However, OSBPL5, a protein of unknown function, was identified as a unique interactor in the NP complex of NY/2009 (Fig. 1d). The PB1 and PB2 interactomes shared several common interactors, such as PRKAR1A and UBR5 (Supplementary Fig. 5a and b). Interestingly, PB1 was associated with DNAJC7, whereas PB2 interacted with DNAJB3 or/and DNAJB6. The M2 interactomes included AGTRAP and EIF2B family members (Supplementary Fig. 5c). Thus, IAVs have common and distinct interactions with the cellular protein network that may underlie the different outcomes of viral pathogenesis.

**Identification of host factors regulating viral infection**. To identify common host factors that modulate viral infection of different IAV strains, we established a 'core' interactome comprising 185 host factors associated with $\geq 3$ strains (Fig. 2a). We next examined the effects of 34 common interactors on viral infection by overexpression in HEK293 cells. After 24 h of transfection, cells were infected with 0.1 MOI of a reporter virus, the PR8 IAV carrying a Gaussia luciferase gene (PR8-Gluc)[23] for 16 h. We determined viral infection activity by luciferase assays (Fig. 2b and Supplementary Data 9). IAV infection was facilitated by EIF2B4 (Fig. 2b). More importantly, four novel proteins (FKBP8, TRIM41, PKP2 and ZMPSTE24) inhibited influenza viral infection without pronounced impact on cell viability (Fig. 2b and Supplementary Fig. 6a). The interactions between these host factors and viral proteins were confirmed by co-immunoprecipitation (Supplementary Fig. 6b–f). RNA interference (RNAi) depletion confirmed the requirement of EIF2B4 for viral replication in A549 and primary tracheal cells (Fig. 2c). Furthermore, RNAi also confirmed the antiviral roles of PKP2, FKBP8, TRIM41 and ZMPSTE24 in A549 cells and primary tracheal cells (Fig. 2c). Consistently, PKP2, FKBP8, TRIM41 and ZMPSTE24 inhibited viral infection of NY/2009 and WSN/33 (Supplementary Fig. 7a). RNAi knockdown efficiency and cell viability were verified (Supplementary Fig. 7b–c). Taken together, we identified five HCIPs that regulate viral infection.

**PKP2 interacts with PB1 polymerase**. PKP2 is a scaffold protein for desmosomal cell–cell junctions, but the role of PKP2 in controlling IAV infection is unknown. Thus, we examined the mechanism of PKP2-mediated viral restriction. First, we examined the interaction of PB1 with endogenous PKP2. Primary human tracheal epithelial cells were infected with PR8 IAV and then lysed for immunoprecipitation. Following PR8 IAV infection, endogenous PKP2 bound to viral PB1 (Fig. 3a). Next, we examined PKP2-PB1 co-localisation. A549 lung epithelial cells were infected with PR8 IAV. Consistent with a previous report[9], endogenous PKP2 was expressed in cell junctions and nuclei. Following infection, PB1 was expressed and co-localized with PKP2 in the nucleus (Fig. 3b). Finally, we determined which domains were required for the PB1 and PKP2 interaction. The PB1 C-terminal region consisting of residues 493 to 757 was sufficient for the interaction with PKP2 (Fig. 3c), whereas the N-terminal domain of PKP2 was required for the interaction with PB1 (Fig. 3d).

**PKP2 restricts IAV infection**. These initial observations led to our working hypothesis that PKP2 may affect influenza viral replication. To test this hypothesis, four assay systems (reporter assay, western blot, immunofluorescence and plaque assay) were used to evaluate the biological effects of PKP2 overexpression on IAV replication. First, three FLAG-tagged armadillo repeat-containing proteins, PKP2, PKP4 (a.k.a. p0071) and β-catenin (also known as CTNNB1) were transfected into HEK293 cells, followed by infection with the PR8-Gluc. PKP2 but not the other armadillo repeat-containing proteins restricted IAV replication (Fig. 4a). PKP2 exhibited minimal effects on the activity of a Wnt signalling reporter (Supplementary Fig. 8). Next, we examined whether PKP2 restricts other RNA viruses. HEK293 cells expressing PKP2-FLAG were infected with IAV, vesicular stomatitis virus or Sendai virus. PKP2 selectively restricted IAV replication (Fig. 4b). We then determined the effect of PKP2 on viral infection by examining the NP protein expression of WSN/33 IAV. PKP2 overexpression resulted in viral restriction, as indicated by decreased levels of NP protein detected by western blot (Fig. 4c). Immunofluorescence assays were also performed to visualize viral restriction. Ectopic expression of PKP2 restricted IAV infection of HEK293 cells for multiple H1N1 strains (PR8, WSN/33, NY/2009) and one H3N2 strain (Aichi) (Fig. 4d,e). Plaque assays were used to determine the effect of PKP2 on the production of infectious IAV particles. Overexpression of PKP2 consistently reduced viral titres in A549 lung epithelial cells (Fig. 4f). Taken together, these findings indicate that PKP2 is an IAV restriction factor.

To complement the above overexpression data, we depleted PKP2 with siRNA. Decreased PKP2 protein expression was correlated with increased IAV reporter activity in A549 lung cells (Fig. 5a and Supplementary Fig. 9a). RNAi depletion of other PKP proteins or CTNNB1 had marginal effects on IAV infection (Supplementary Fig. 9b), indicating the specificity of PKP2 on IAV restriction. To exclude off-target effects of RNAi, cells were transfected with a siRNA resistant PKP2 rescue construct before infection with PR8 reporter virus. When combined with PKP2 siRNA the rescue construct restored antiviral activity, validating siRNA specificity (Fig. 5b). In addition, silencing PKP2 enhanced virus propagation of four IAV strains in A549 cells, as detected by immunofluorescence assays (Fig. 5c). Knockdown of PKP2 also enhanced IAV propagation in primary tracheal epithelial cells, as detected by plaque assay (Fig. 5d). Therefore, endogenous PKP2 is essential for host restriction to IAV.

**PKP2 inhibits IAV polymerase activity**. Heterotrimeric IAV polymerase requires interactions between PB2 and the C-terminus of PB1. PKP2 also interacts with the PB1 C-terminal domain (Fig. 3c). These findings led us to examine whether PKP2 can compete with PB2 for PB1 binding, thereby limiting IAV polymerase activity. Indeed, PKP2 competed with PB2 and reduced $\sim 70\%$ of PB2 protein in the PB1 complex (Fig. 6a), whereas the PB1–PA and PB1–NP interactions remained intact (Fig. 6b,c). We also found that NP, PB2 and PA failed to pull down PKP2, suggesting the interaction specificity between PB1 and PKP2 (Supplementary Fig. 10a). To investigate the effect of PKP2 on the IAV polymerase complex, we first infected HEK293 cells stably expressing PKP2-FLAG or GFP-FLAG with 1 MOI of PR8 IAV for 16 h. Then, we fractionated the cell lysates by centrifugation over a 15–55% sucrose gradient. Notably, PKP2 co-fractionated with the majority of IAV polymerase complex distributed from fractions #4 to #7, whereas green fluorescent protein was absent in these fractions (Fig. 6d). Furthermore, PKP2 perturbed the fraction distribution of IAV polymerase complex and reduced the PB2/PB1 protein ratio in fractions

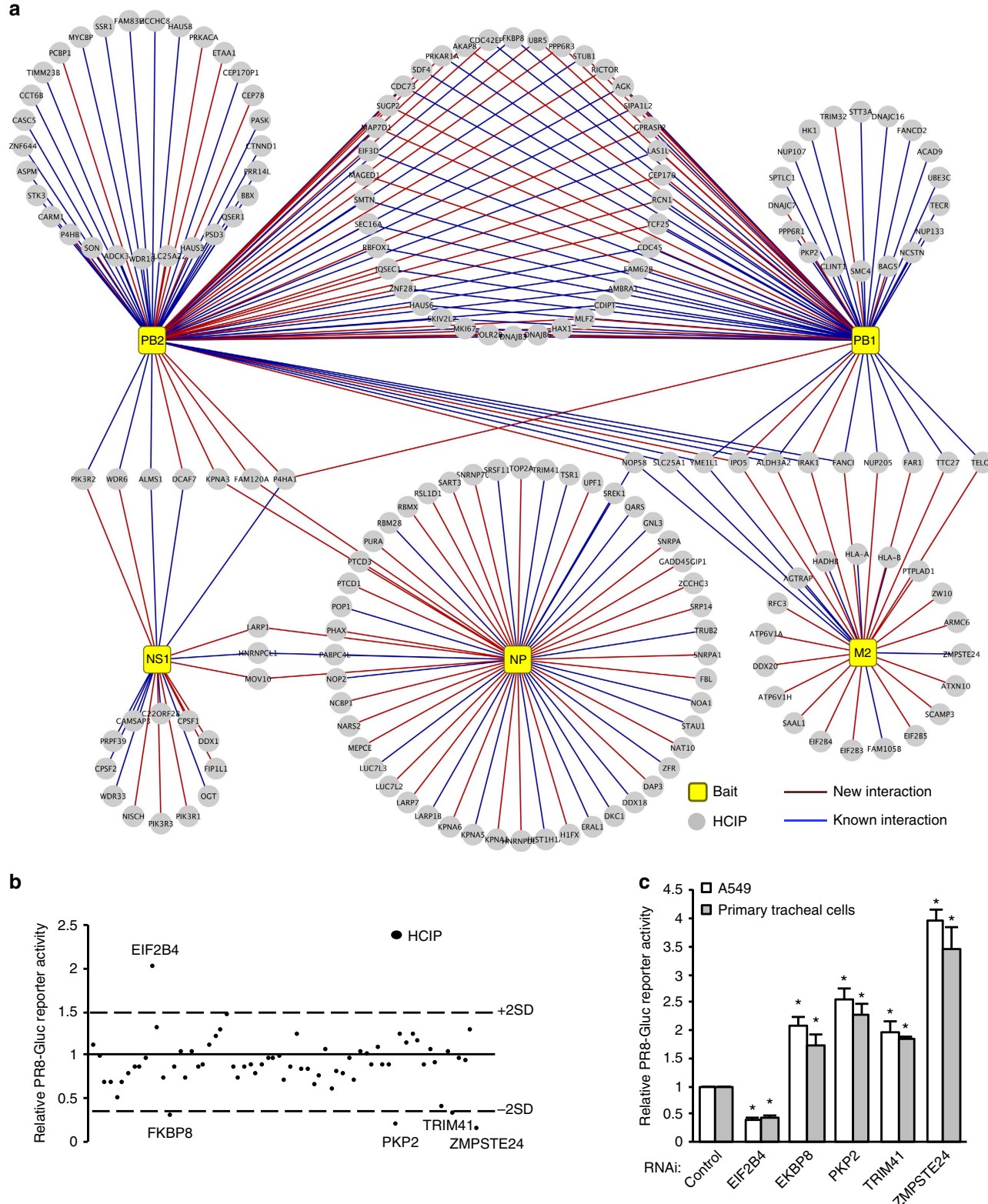

**Figure 2 | Identifying HCIPs that regulate IAV infection by screening the core interactome.** (**a**) Map of the core IAV–host protein interactome that comprises HCIPs associated with multiple IAV strains in Fig. 1b. Legends are indicated. (**b**) PR8-Gluc reporter screening assay in HEK293 cells. At 24 h after transfection with HCIPs, the cells were infected with 0.1 MOI of PR8-Gluc. After 16 h, IAV infection activity was determined by luciferase activity. The screenings were biologically repeated two times. The dash lines indicate $\pm 2 \times$ s.d. of the whole screening data set. The HCIPs with $> 2 \times$ s.d. or $< -2 \times$ s.d. are labelled. (**c**) A549 and primary human tracheal epithelial cells were transfected with control siRNA or the indicated siRNA duplexes. After 48 h, the cells infected with 0.1 MOI of PR8-Gluc for 16 h. The relative luciferase activity was examined. Data represent means $\pm$ s.d. of three independent experiments. The $P$ value was calculated (two-tailed Student's $t$-test) by comparison with the corresponding siRNA control in each cell group. An asterisk indicates $P < 0.05$.

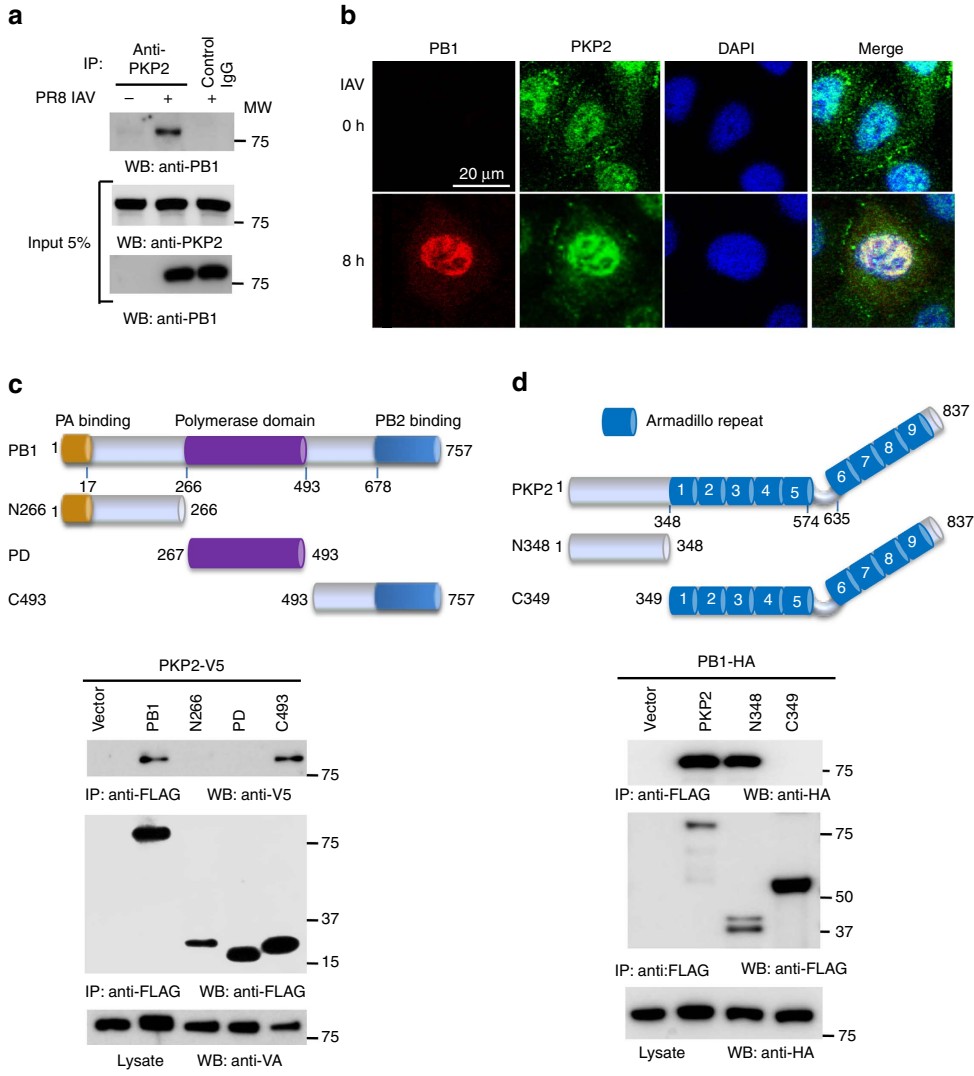

**Figure 3 | PKP2 interacts and co-localizes with PB1. (a)** Primary human tracheal epithelial cells were infected with 1 MOI of PR8 IAV for 16 h. Cell lysates were subjected to immunoprecipitation and immunoblotting with the indicated antibodies to detect endogenous interactions. Molecular weights (MW) are indicated. **(b)** A549 cells were infected with 1 MOI of PR8 IAV for 8 h and stained with anti-PKP2 (green), anti-PB1 (red) and DAPI nuclear stain (blue). **(c)** Full-length and various PB1 deletion mutants were fused with FLAG epitope and co-transfected with PKP2-V5 into HEK293 cells. Lysates were immunoprecipitated with anti-FLAG antibody and blotted with the indicated antibodies. **(d)** Full-length and various PKP2 deletion mutants were fused with FLAG epitope and co-transfected with PB1-HA into HEK293 cells. Lysates were immunoprecipitated with anti-FLAG antibody and blotted with the indicated antibodies.

containing PKP2 (Fig. 6d). Finally, we examined whether PKP2 had an effect on PB1 polymerase activity using a polymerase reporter assay. Silencing PKP2 with siRNA enhanced polymerase activity, whereas overexpression of PKP2 reduced polymerase activity (Fig. 6e and Supplementary Fig. 10b). Thus, PKP2 alters the integrity of IAV polymerase complex and impairs IAV replication.

## Discussion

We constructed and comparatively analysed several IAV–host protein interactomes. As with any screening approach, the database does not represent a complete interaction network and many of the hits identified in this screen require additional validation using more directed experiments. Nonetheless, the comparative interactomes provide insights on the virulence and host determinants. For example, the various NS1 proteins exhibit different binding abilities to CPSF proteins and the CPSF-binding partners FIP1L1 and WDR33 (ref. 24). NS1 interacts with CPSF

and inhibits the cleavage and polyadenylation of host pre-mRNA[25]. In our network, CPSF, FIP1L1 and WDR33 strongly interact with NS1 of WSN/33 and Aichi, but not with PR8 NS1. NS1 of PR8 influenza lacks F103 and M106 residues that are required to stabilize the CPSF interaction[21]. Notably, NS1 of NY/2009 weakly interacts with CPSF even though it possesses intact F103 and M106. Dr Garcia-Sastre's group also suggest that other binding sites are also required[26]. In addition to CPSF, many new strain-specific NS1 interactors were identified, including FKBP5, NKRF, TNFAIP8, ZFC3H1 and TRNAU1AP. FKBP5, NKRF and TNFAIP8 are involved in NF-κB signalling[27–29]. ZFC3H1 was recently shown to be required for IL-8 production[30]. These unique interactions suggest distinct perturbation strategies of each IAV strain, potentially enabling the definition of *de novo* virulence and host determinants.

Systematic comparisons of strain-specific influenza–host protein interactomes are also essential for the identification of general mechanisms of regulating viral infection and the

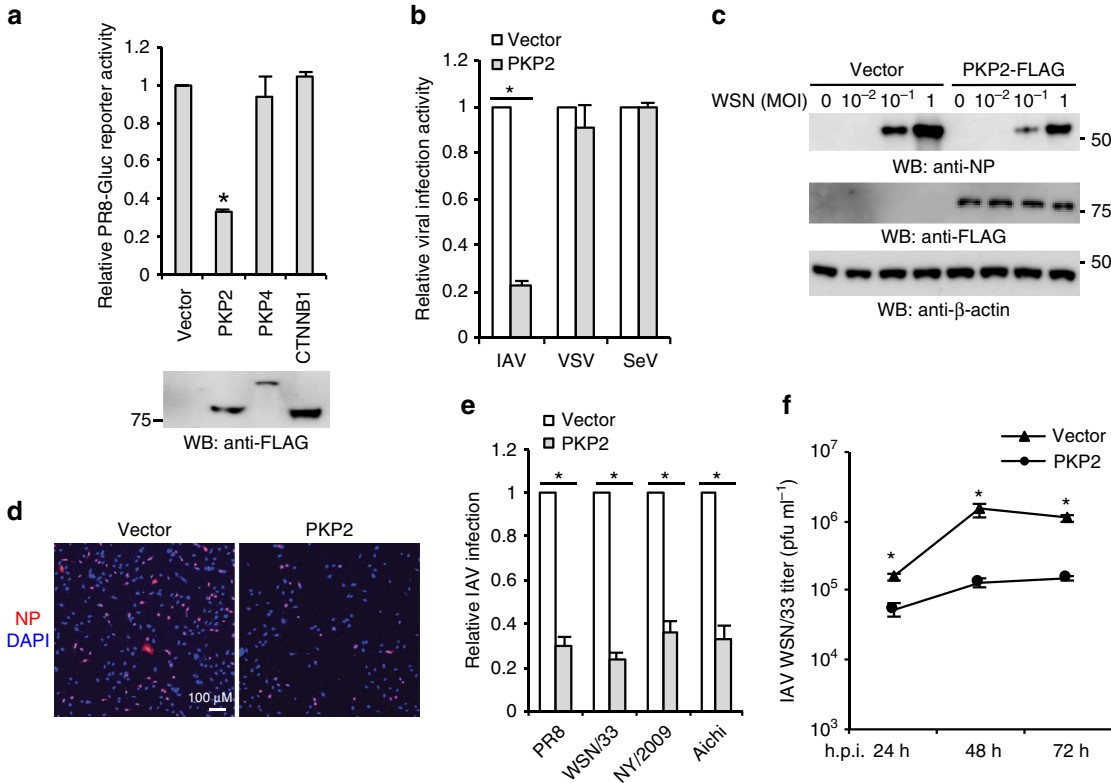

**Figure 4 | PKP2 restricts IAV infection.** (**a**) HEK293 cells were transfected with pCMV3-tag-8 vector, PKP2-FLAG, PKP4-FLAG or CTNNB1 (β-catenin)-FLAG for 24 h, followed by infection with 0.1 MOI of PR8-Gluc for 16 h. Luciferase activity assays were performed. All experiments were biologically repeated three times. Data represent means ± s.d. of three independent experiments. The *P* value was calculated (two-tailed Student's *t*-test) by comparison with the vector control. An asterisk indicates *P* < 0.05. Western blot shows the protein expression levels from one experiment. (**b**) HEK293 cells transfected with PKP2-FLAG or pCMV3-tag-8 vector were infected with 0.1 MOI of IAV-Gluc, VSV-Luc and SeV-Luc for 16 h. Data represent means ± s.d. of three independent experiments. The *P* value was calculated (two-tailed Student's *t*-test) by comparison with the vector control. An asterisk indicates *P* < 0.05. (**c**) HEK293 cells were transfected with pCMV3-tag-8 vector or PKP2-FLAG for 24 h, then infected with WSN/33 IAV for 16 h. Cell lysates were blotted using the indicated antibodies. (**d**) HEK293 cells were transfected with pCMV3-tag-8 vector or PKP2-FLAG for 24 h, then infected with WSN/33 IAV for 16 h. Fixed cells were stained with an anti-NP antibody. The percentage of stained cells is summarized in **e**. (**e**) HEK293 cells were transfected with the pCMV3-tag-8 vector or PKP2-FLAG for 24 h, then infected with 1 MOI of PR8, WSN/33, NY/2009 or Aichi IAV for 16 h. Fixed cells were stained with anti-NP antibody (PR8 and WSN/33), anti-HA (NY/2009) or anti-Aichi antibodies. The relative infection was determined by the ratio of positive cells. Data represent means ± s.d. of three independent experiments (>80 cells counted per experiment). The *P* value was calculated (two-tailed Student's *t*-test) by comparison with the pCMV3-tag-8 vector control. An asterisk indicates *P* < 0.05. (**f**) A549 cells were transfected with PKP2-FLAG. After 48 h, cells were infected with 0.001 MOI of WSN/33 IAV. After the designated hour post infection (h.p.i.), virus titers were determined by plaque assay. All experiments were biologically repeated three times. Data represent means ± s.d. of three independent experiments. The *P* value was calculated (two-tailed Student's *t*-test) by comparison with the pCMV3-tag-8 vector control. An asterisk indicates *P* < 0.05.

discovery of common therapeutic targets. For example, IPO5 is a major common interactor of PB1 proteins and required for IAV polymerase nuclear import[19]. The heat shock protein Hsp90 is a host factor facilitating viral RNA synthesis[31]. In our IAV–host interactomes, the heat shock proteins DNAJCs are common interactors for PB1 and PB2. Whether these heat shock proteins regulate viral RNA synthesis requires further investigation. By functional screening the core interactome, we identified four common interactors (ZMPSTE24, FKBP8, PKP2 and TRIM41) that restrict IAV infection and may lead to the discovery of new general antiviral mechanisms.

In the core interactome, PKP2 interacts with influenza PB1 proteins and restricts IAV infection. PKP2, along with PKP1 and PKP3 form a subgroup of catenin protein family that is characterized by the armadillo repeats. PKPs are composed of a basic N-terminal head domain, followed by nine armadillo repeats and a short C-terminal tail[32,33]. A spacer between the fifth and sixth repeat results in a characteristic kink in the domain structure[34]. The N terminus of PKPs is diverse and exhibits no obvious homology. Our study showed that the N terminus of

PKP2 interacted with PB1, suggesting the interaction specificity between PKP2 and PB1. Consistently, PKP2, but not other PKPs restrict IAV infection.

PKPs are scaffold proteins that are essential for the formation of desmosome and stabilisation of cell junctions. PKP2 is the largest and most prevailing plakophilin protein that is expressed in all cell types with desmosomal junctions[9]. Although PKP2 was originally identified as cell junction protein anchoring on plasma membrane, it also localizes in the nucleus[9]. Interestingly, nuclear PKP2 associates with the largest subunit of the RNA polymerase III holoenzyme subunit of transcription factor IIIB (TFIIIB), an enzyme important for transcription of ribosomal RNA and tRNA[35]. However, the biological significance of the interaction with RNA polymerase III is not known.

Our data demonstrate that PKP2 binds to the C-terminal domain of PB1, thus blocking the interaction of PB2 with PB1. Assembly of the heterotrimeric IAV polymerase requires interactions between the PB1 N-terminal domain with the PA chain, whereas the PB1 C-terminus associates with the polymerase PB2 protein[36–38]. The interactions between the

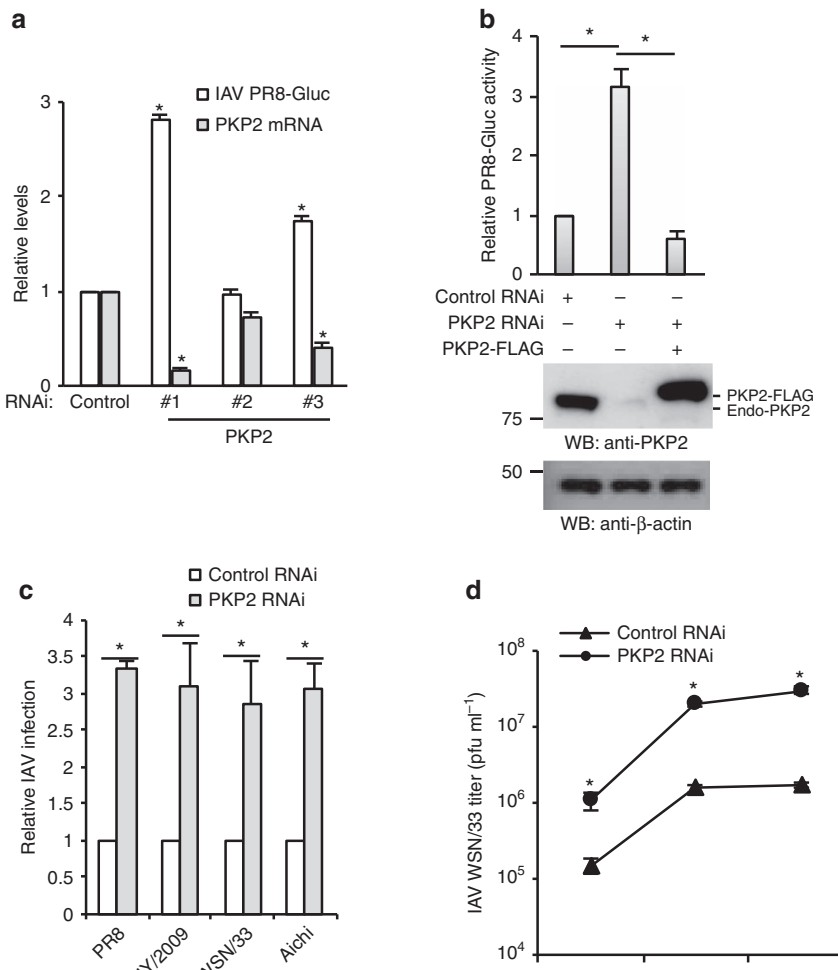

**Figure 5 | RNAi depletion of PKP2 promotes influenza virus infection. (a**) A549 cells were transfected with scrambled control siRNA and 3 siRNA oligos against PKP2. After 48 h, the cells were infected with 0.1 MOI of IAV PR8-Gluc for 16 h. Relative luciferase activities were examined and PKP2 mRNA levels were determined by RT-PCR. All experiments were biologically repeated three times. Data represent means ± s.d. of three independent experiments. The *P* value was calculated (two-tailed Student's *t*-test) by comparison with the siRNA control in each cell group. An asterisk indicates *P* < 0.05. (**b**) HEK293 cells were transfected with siRNA targeting the 3′-UTR of PKP2 (duplex #1) and/or PKP2-FLAG resistant to PKP2 siRNA. After 48 h, the cells were infected with 0.1 MOI of PR8-Gluc for 16 h. The relative luciferase activity is shown. All experiments were biologically repeated three times. Data represent means ± s.d. of three independent experiments. The *P* value was calculated (two-tailed Student's *t*-test) by comparison with the siRNA control. An asterisk indicates *P* < 0.05. Western blot demonstrates the knockdown efficiency. FLAG-tagged PKP2 and endogenous PKP2 (Endo-PKP2) are indicated. (**c**) A549 cells were transfected with PKP2 siRNA (duplex #1) for 48 h, then infected with 1 MOI of PR8, WSN/33, NY/2009 or Aichi IAV. Fixed cells were stained with anti-NP antibody (PR8 and WSN/33), anti-HA (NY/2009) or anti-Aichi antibodies. The relative infection was determined from the ratio of positive cells. Data represent means ± s.d. of three independent experiments (>80 cells counted per experiment). *P < 0.05 significantly different versus control RNAi as calculated by the two-tailed Student's *t*-test. (**d**) Tracheal epithelial cells transfected with control or PKP2 siRNA were infected with 0.001 MOI of WSN/33 IAV for the indicated h.p.i. Titers of culture supernatants containing IAV were determined on MDCK cells, and plaques were enumerated. Data represent means ± s.d. of three independent experiments. The *P* value was calculated (two-tailed Student's *t*-test) by comparison with the control RNAi. An asterisk indicates *P* < 0.05.

polymerase subunits are attractive drug targets because proper assembly of these subunits is crucial for normal polymerase activity. Several inhibitors of synthetic peptides based on the PB1–PB2 interaction domain have been reported to interfere with PB1–PB2 interactions and disrupt normal function of the polymerase[39,40]. Our study suggests that PKP2 is a natural inhibitor of IAV polymerase complex. Defining the critical interaction sites of PKP2 will provide a basis for the development of new antivirals.

In summary, these comparative IAV–host interactomes will contribute to the understanding of host defence mechanisms and might ultimately provide the insights and opportunities necessary for the identification of new antiviral drug targets.

## Methods

**Antibodies.** Anti-β-actin (Abcam, # ab8227, WB (1:1,000)), anti-FLAG (Sigma, # F3165, WB (1:1,000), IFA (1:100)), anti-GFP (Santa Cruz Biotechnology, # sc-9996, WB (1:2,000)), anti-HA epitope (Sigma, # H3663, WB (1:1,000)), anti-NP (BEI resources, # NR-4282, WB (1:1,000), IFA (1:100)), anti-NP (EMD Millipore, # MAB8800, IFA (1:1,000)), anti-HA from A/California/04/2009 (H1N1) (BEI resources, # NR-42021, IFA (1:100)), anti-influenza A/Aichi/2/68 (BEI resources, # NR-3125, IFA (1:100)), anti-PB1 (BEI resources, # NR-31690, WB (1:1,000)), anti-PB2 (Genetex, # GTX125926, WB (1:1,000)), anti-PA (Genetex, # GTX125932, WB (1:1,000)), anti-PKP2 (Fitzgerald Industries International, # 20R-2656, WB (1:1,000), IFA (1:100)), anti-CPSF4 (Millipore, # MABE620, WB (1:1,000)).

Goat anti-Mouse IgG-HRP (Santa Cruz Biotechnology, # sc-2055, WB (1:10,000)), Goat anti-Rabbit IgG-HRP (Santa Cruz Biotechnology, # sc-2030, WB (1:10,000)), Goat anti-Guinea Pig IgG-HRP (Thermo Fisher Scientific, # A18775, WB (1:10,000)), Alexa Fluor 594 Goat Anti-Mouse IgG (H + L) (Life Technologies, # A11005, IFA (1:200)), Alexa Fluor 488 Goat Anti-Rabbit IgG (H + L)

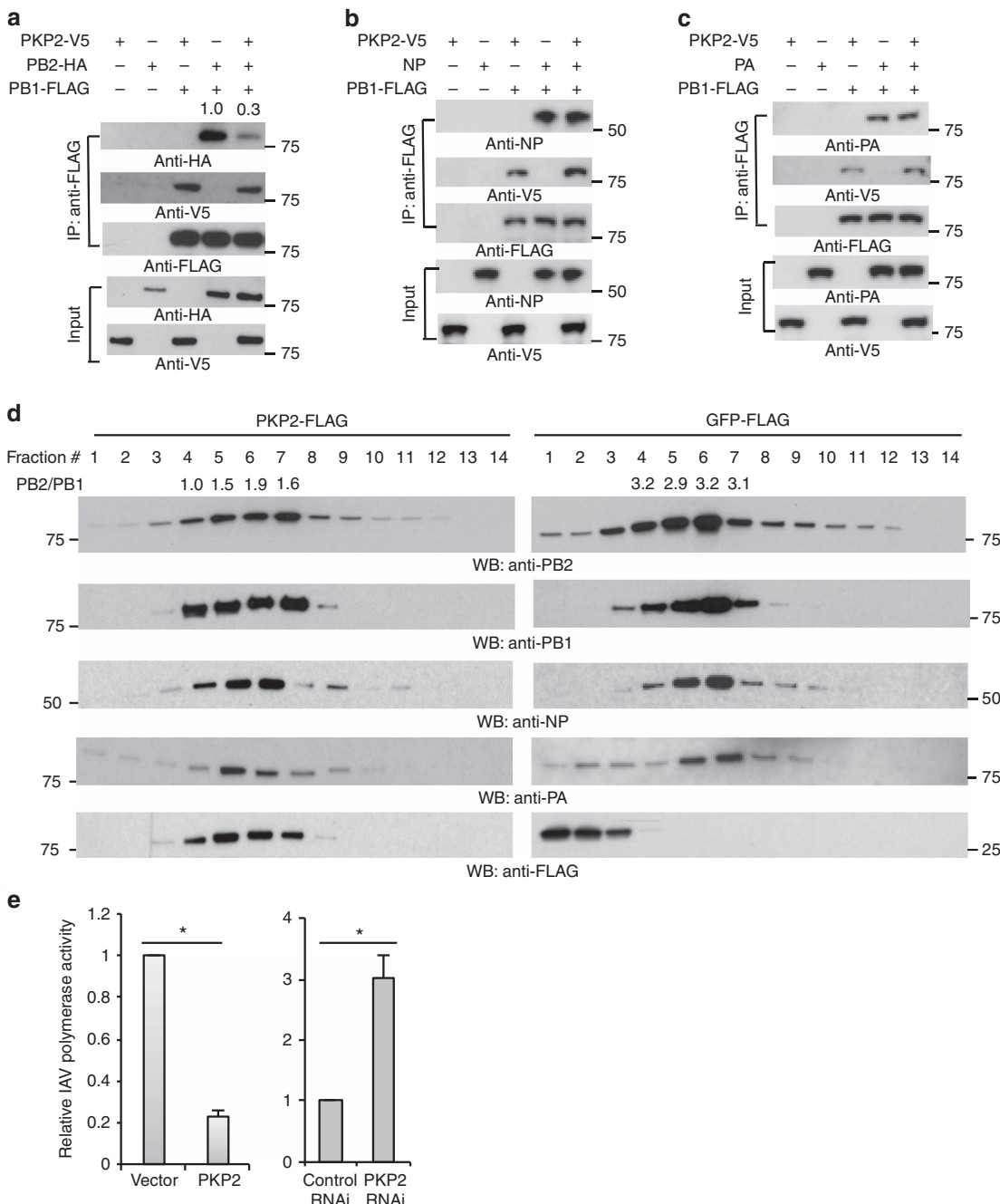

**Figure 6 | PKP2 perturbs the IAV polymerase complex and inhibits viral polymerase activity. (a)** PB1 and PB2 were co-transfected with PKP2 into HEK293 cells. After 48 h, cell lysates were harvested for immunoprecipitation using anti-FLAG antibody and blotted using the indicated antibodies. **(b)** PB1 and NP were co-transfected along with PKP2 into HEK293 cells. After 48 h, cell lysates were harvested for immunoprecipitation using anti-FLAG antibody. The indicated antibodies were used for blotting. **(c)**. PB1 and PA were co-transfected with PKP2 into HEK293 cells. After 48 h, cell lysates were harvested for immunoprecipitation using anti-FLAG antibody and blotted as indicated. **(d)** HEK293 cells stably expressing PKP2-FLAG or GFP-FLAG were infected with 1 MOI of PR8 IAV for 16 h. Then the cell lysates were separated by 15–55% sucrose density centrifugation. Fractions were blotted using the indicated antibodies. The ratios of PB2 to PB1 in the fractions 4–7 were indicated. **(e)** HEK293 cells were transiently transfected with a plasmid cocktail containing PB1, PB2, PA, NP expression plasmids of PR8 IAV plus a polymerase I plasmid expressing an influenza virus-like RNA encoding the reporter protein firefly luciferase, along with control siRNA, PKP2 siRNA, the vector pCMV3-tag-8 or PKP2-FLAG for 48 h. The relative luciferase signal is shown. The transfection efficiency was determined by western blot (Supplementary Fig. 10b). Data represent means ± s.d. of three independent experiments. The *P* value was calculated (two-tailed Student's *t*-test) by comparison with the corresponding control. An asterisk indicates $P < 0.05$.

(Life Technologies, # A11034, IFA (1:200)), Alexa Fluor 488 Goat Anti-Guinea Pig IgG (H + L) (Life Technologies, # A11073, IFA (1:200)).

**Plasmids.** PR8 IAV viral genes are kind gift from Dr Ervin Fodor (University of Oxford, Oxford, UK)[41]. WSN/33 IAV viral genes are generous gift from Dr Robert Webster (St Jude Children's Hospital, Memphis, TN)[42]. NY/2009 IAV viral genes are generous gift from J. Craig Venter Institute (La Jolla, CA). NS1 of VN/2004 IAV is a kind gift from Dr Andrew Rice (Baylor College of Medicine, TX). The NP of Aichi IAV (NCBI, # AFM71861.1) was ordered from Sino Biological Inc., Beijing, China (# VG40207-UT). The PB1, PB2 and PA of VN/2004 IAV are generous gift from Koyu Hara (Kurume University School of Medicine, Japan)[43].

The M2 of VN/2004 IAV (NCBI, # EF541452) and NS1 of Aichi IAV (NCBI, # AFM71862.1) were synthesized by GenScript (Piscataway, NJ). Various PB1 mutants of IAV PR8 were constructed using a QuikChange II Site-Directed Mutagenesis kit (Agilent Technologies, # 200523).

Human PKP2-FLAG was a gift from Kathleen Green (Addgene plasmid # 32230)[44]. Point mutations and deletions of PKP2-FLAG were constructed using a QuikChange II Site-Directed Mutagenesis kit. Human PKP4-FLAG (a.k.a p0071-FLAG) was a gift from Dr Andrew Kowalczyk (Emory University, Atlanta, GA)[45]. Human EIF2B4-V5 was purchased from Harvard PlasmID Database (# HsCD00414983). Human TRIM41-HA was a generous gift from Dr Adolfo Garcia-Sastre (Mount Sinai School of Medicine, NY)[46]. Human ZMPSTE24 cDNA (Harvard PlasmID Database, # HsCD00075979) was cloned into pCMV-3Tag-8 (Agilent Technologies, # 240203) to make ZMPSTE24-FLAG. Human FKBP8 cNDA (Harvard PlasmID Database, # HsCD00082457) was cloned into pCMV-3Tag-8 to make FKBP8-FLAG. CTNNB1-FLAG was from Dr He laboratory as described[47]. The Wnt signalling reporter, TOPFLASH was purchased from Upstate Biotechnology (#21–170). The control plasmid, pCMV-3Tag-8 was purchased from Agilent Technologies (#240203).

**Cells.** HEK293 cells (ATCC, # CRL-1573) and MDCK cells (ATCC, # CCL-34) were maintained in Dulbecco's Modified Eagle Medium (Life Technologies, # 11995-065) containing antibiotics (Life Technologies, # 15140-122) and 10% fetal bovine serum (Life Technologies, # 26140-079). A549 cells (ATCC, # CCL-185) were cultured in RPMI Medium 1640 (Life Technologies, # 11875-093) plus 10% fetal bovine serum and $1 \times$ MEM Non-Essential Amino Acids Solution (Life Technologies, # 11140-050). Primary human tracheal epithelial cells and supporting medium were purchased from Lifeline Technology (Frederick, MD) (# FC-0035 and # LL-0023). The cell lines were examined using MycoAlert Mycoplasma Detection Kit (Lonza, # LT07-418).

**Viruses.** Influenza A/Puerto Rico/8/34 (H1N1) (Charles River Laboratories, Wilmington, MA, # 10100374), influenza A/WSN/33 (H1N1) (a kind gift of Dr Peter Palese, Mount Sinai School of Medicine, NY), influenza A virus A/New York/18/2009 (H1N1) pdm09 (BEI Resources, # NR-15268), and A/Aichi/68 (H3N2) (Charles River Laboratories, # 10100375). Influenza PR8-GLuc virus was a generous gift from Dr Peter Palese and features a *Gaussia* luciferase (Gluc) gene inserted downstream of PB2 (ref. 48). IAV was propagated in specific pathogen-free fertilized eggs Premium Plus (Charles River Laboratories) as described by Szretter et al.[49] In total, 9–11-day-old embryonated chicken eggs are used for the production of influenza virus. 0.2 ml stock influenza virus at $1 \times 10^3$ TCID50 was injected through the puncture hole into the allantoic cavity. After 72 h of incubation, allantoic fluid was collected. IAV titers were determined by plaque assay as described by Matrosovich et al.[50] In brief, $1.2 \times 10^6$ MDCK cells per ml were split into six-well plates. After $2 \times$ washes with Dulbecco's Modified Eagle Medium, serial dilutions of IAV were adsorbed onto the cells for 1 h. The cells were covered with Dulbecco's Modified Eagle Medium containing $1 \times$ Avicel RC591 NF (FMC Biopolymer, Philadelphia, PA) and $1 \, \mu g \, ml^{-1}$ TPCK-trypsin (Thermo Fisher Scientific, # 20233). Crystal violet staining was performed 48 hour post infection , and visible plaques were counted.

Sendai virus with luciferase was a kind gift from Dr Charles Russell (St. Jude's Hospital, Memphis, TN). Vesicular stomatitis virus with firefly luciferase gene (VSV-Luc) was a kind gift from Dr Sean Whelan (Harvard Medical School, Boston, MA).

**Sample preparation, western blot and immunoprecipitation.** Approximately $1 \times 10^6$ cells were lysed in 500 μl of tandem affinity purification lysis buffer (50 mM Tris-HCl (pH 7.5), 10 mM MgCl$_2$, 100 mM NaCl, 0.5% Nonidet P40, 10% glycerol, Complete EDTA-free protease inhibitor cocktail tablets (Roche, # 11873580001)) for 30 min at 4 °C. The lysates were then centrifuged for 30 min at 15,000g. Supernatants were collected and mixed with $1 \times$ Lane Marker Reducing Sample Buffer (Thermo Fisher Scientific, # 39000).

Precision Plus Protein Dual Color Standards (5 μl) (Bio-Rad, # 161-0374) and samples (10–15 μl) were loaded into Mini-Protean TGX Precast Gels, 15 well (Bio-Rad, # 456-103), and run in $1 \times$ Tris/Glycine/SDS Buffer (Bio-Rad, # 161-0732) for 35 min at 200 V. Protein samples were transferred to Immun-Blot PVDF Membranes (Bio-Rad, # 162-0177) in $1 \times$ Tris/Glycine Buffer (Bio-Rad, # 161-0734) at 70 V for 60 min. PVDF membranes were blocked in $1 \times$ TBS buffer (Bio-Rad, # 170-6435) containing 5% Blotting-Grade Blocker (Bio-Rad, # 170-6404) for 1 h. After washing with $1 \times$ TBS buffer for 30 min, the membrane blot was incubated with appropriately diluted primary antibody in antibody dilution buffer ($1 \times$ TBS, 5% BSA, 0.05% sodium azide) at 4 °C for 16 h. Then, the blot was washed three times with $1 \times$ TBS (each time for 10 min) and incubated with secondary HRP-conjugated antibody in antibody dilution buffer (1:10000 dilution) at room temperature for 1 h. After three washes with $1 \times$ TBS (each time for 10 min), the blot was incubated with Clarity Western ECL Substrate (Bio-Rad, # 170-5060) for 1–2 min. The membrane was removed from the substrates, then exposed either to HyBlot CL Autoradiography Film (Denville Scientific Inc. # E3018) in the dark room or to Amersham imager 600 (GE Healthcare Life

Sciences, Marlborough, MA). Uncropped scans of western blots are provided in Supplementary Fig. 11.

For immunoprecipitation, 2% of cell lysates were saved as an input control, and the remainder was incubated with 5–10 μl of the indicated antibody plus 20 μl of Pierce Protein A/G Plus Agarose (Thermo Fisher Scientific, # 20423), 10 μl of EZview Red Anti-FLAG M2 Affinity Gel (Sigma, # F2426), or 10 μl of Anti-HA-Agarose (Sigma, # A2095). After mixing end-over-end at 4 °C overnight, the beads were washed 3 times (1 min each wash) with 500 μl of lysis buffer. Anti-HA Agarose and Anti-FLAG M2 Affinity Gel were eluted with 50 μl of 1 mg ml$^{-1}$ HA peptide (Sigma, # I1249) or 5 mg ml$^{-1}$ 3X FLAG peptide (Sigma, # F4799), respectively. Immunoprecipitated complexes with Protein A/G Plus Agarose were mixed with $1 \times$ Lane Marker Reducing Sample Buffer.

**Immunofluorescence assay.** Cells were cultured in the Lab-Tek II CC2 Chamber Slide System 4-well (Thermo Fisher Scientific, # 154917). After the indicated treatment, the cells were fixed and permeabilized in cold methanol for 10 min at $-20$ °C. Then, the slides were washed with $1 \times$ PBS for 10 min and blocked with Odyssey Blocking Buffer (LI-COR Biosciences, # 927-40000) for 1 h. The slides were incubated in Odyssey Blocking Buffer with appropriately diluted primary antibodies at 4 °C for 16 h. After 3 washes (5 min per wash) with $1 \times$ PBS, the cells were incubated with the corresponding Alexa Fluor conjugated secondary antibodies (Life Technologies) for 1 h at room temperature. The slides were washed three times (5 min each time) with $1 \times$ PBS and counterstained with 300 nM DAPI for 1 min, followed by washing with $1 \times$ PBS for 1 min. After air-drying, the slides were sealed with Gold Seal Cover Glass (Electron Microscopy Sciences, # 3223) using Fluorogel (Electron Microscopy Sciences, # 17985-10). Images were captured and analysed using an Olympus Fluoview Confocal microscope (Olympus, Center Valley, PA).

**Real-time PCR.** Total RNA was prepared using RNeasy columns (Qiagen, # 74106). In total, 1 μg quantity of RNA was transcribed into cDNA using a QuantiTect reverse transcription kit (Qiagen, # 205311). For one real-time reaction, 20 μl of SYBR Green PCR reaction mix (Roche Applied Science) including a 1/10 volume of the synthesized cDNA plus an appropriate oligonucleotide primer pair were analysed on a LightCycler 480 (Roche). The comparative Ct method was used to determine the relative mRNA expression of genes normalized by the house-keeping gene GAPDH. The primer sequences: PKP2, forward primer 5′-GTGGG CAACGGAAATCTTCAC-3′, reverse primer 5′-CCAGCCTTTAGCATGTCAT AGG-3′; GAPDH, forward primer 5′-AGGTGAAGGTCGGAGTCA-3′, reverse primer 5′-GGTCATTGATGGCAACAA-3′.

**Plasmid transfection.** HEK293 cells were transfected using Lipofectamine 2000 Transfection Reagent, and A549 cells were transfected using Lipofectamine 3000 Transfection Reagent (Life Technologies, # L3000015) according to the manufacturer's protocol. In total, 0.5 μg HCIP plasmid was used for overexpression experiments.

**RNAi depletion.** RNAi target sequences (sense strand): PKP2 siRNA #1: 5′- ACG GCTCATGTTAATGAGTTA-3′; PKP2 siRNA #2: 5′- TCCGTGGGCAACGGAAA TCTT-3′; PKP2 siRNA #3: 5′-ATCCAGCGAAATGAATCTACA-3′. ZMPSTE24 siRNA: 5′-CAATCTATGCTGATTATAT-3′. EIF2B4 FlexiTube siRNA (Qiagen, # GS8890). TRIM41 FlexiTube siRNA (Qiagen, # GS90933). FKBP8 FlexiTube siRNA (Qiagen, # GS23770). siGENOME Non-Targeting Control siRNA (Dharmacon, # D-001210-02-05) was used as the control siRNA. siRNA duplexes were transfected into A549 (5 pmol) and primary tracheal cells (5 pmol) using Lipofectamine RNAiMAX Transfection Reagent (Life Technologies, # 13778030) according to the manufacturer's protocol.

**Sucrose gradient centrifugation.** A total of $1 \times 10^8$ HEK293 cells stably expressing FLAG-tagged PKP2 were collected and lysed in gradient centrifugation (GC) buffer (25 mM Tris·HCl (pH 7.5), 150 mM KCl, 0.5% Nonidet P40, 2 mM EDTA, 1 mM Sodium fluoride, 0.5 mM dithiothreitol, phosphatase inhibitors, and protease inhibitors). The lysates were centrifuged at 16,000g for 30 min at 4 °C to remove cell debris. The lysates were then separated in GC buffer with 15% (wt vol$^{-1}$) to 55% (wt vol$^{-1}$) sucrose by ultracentrifugation in an SW41 Ti rotor (Beckman Coulter, Inc.) at 111,000g for 18 h at 4 °C. Fractions were collected and loaded onto SDS-PAGE gels for western blotting.

**Reconstitution of influenza virus polymerase activity.** HEK293 cells were transfected with vectors expressing PR8 PB1(0.05 μg), PB2 (0.05 μg), NP (0.05 μg), PA (0.05 μg) and the indicated PKP2 (0.2 μg) or siPKP2 duplex (5 pmol) in addition to a polymerase I (PolI)-driven plasmid transcribing an IAV-like RNA encoding the reporter protein firefly luciferase (0.1 μg) to monitor viral polymerase activity[40]. Cells were lysed 48 h after transfection. Luciferase activity was measured with a luciferase assay system (Promega, Madison, WI). Transfection efficiency was monitored by western blot.

**Cell viability.** Cell viability was assessed by using the CellTiter-Glo Luminescent Cell Viability Assay (Promega, # G7570) according to the manufacturer's instructions.

**Stable cell line selection.** Two mirograms of each viral construct was transfected into HEK293 cells using Lipofectamine 2000 (Invitrogen, # 11668027). Two days after transfection cells were treated with 3 μg ml$^{-1}$ puromycin or 100 μg ml$^{-1}$ hydromycin for 14 days. Single colonies were picked and expanded in six-well plates. Protein expression levels in each colony were determined by immunoblotting.

**Purification of protein complexes and mass spectrometry.** AP-MS experiments were performed as previously described[12]. For protein purification, HEK293 cell lines stably expressing each FLAG-tagged viral protein were divided into two groups. Then, one group was infected with 1 MOI of IAV. After 16 h of infection, $10^8$ cells were collected and lysed in 10 ml of tandem affinity purification buffer[4]. Cell lysates were precleared with 50 μl of protein A/G resin before the addition of 20 μl of anti-FLAG resin (Sigma, # F2426) and incubation for 16 h at 4 °C on a rotator. The resin was washed three times and transferred to a spin column with 40 μl of 3X FLAG peptide for 1 h at 4 °C on a rotator. The purified complexes were loaded onto a 4–15% NuPAGE gel. The gels were stained with a SilverQuest staining kit (Invitrogen), and lanes were excised for mass spectrometry analysis by the Taplin Biological Mass Spectrometry Facility (Harvard Medical School, Boston, MA).

**Mass spectrometry.** Excised gel bands were cut into ∼1 mm$^3$ pieces. Gel pieces were then subjected to a modified in-gel trypsin digestion procedure. Gel pieces were washed and dehydrated with acetonitrile for 10 min. followed by removal of acetonitrile. Pieces were then completely dried in a speed-vac. Gel pieces were rehydrated with 50 mM ammonium bicarbonate solution containing 12.5 ng μl$^{-1}$ modified sequencing-grade trypsin (Promega, Madison, WI) at 4 °C. After 45 min., the excess trypsin solution was removed and replaced with 50 mM ammonium bicarbonate solution to just cover the gel pieces. Peptides were later extracted by removing the ammonium bicarbonate solution, followed by one wash with a solution containing 50% acetonitrile and 1% formic acid. The extracts were then dried in a speed-vac (∼1 h) and stored at 4 °C until analysis.

On the day of analysis the samples were reconstituted in 5–10 μl of HPLC solvent A (2.5% acetonitrile, 0.1% formic acid). A nano-scale reverse-phase HPLC capillary column was created by packing 5 μm C18 spherical silica beads into a fused silica capillary (100 μm inner diameter × ∼12 cm length) with a flame-drawn tip. After equilibrating the column each sample was loaded via a Famos auto sampler (LC Packings, San Francisco CA) onto the column. A gradient was formed and peptides were eluted with increasing concentrations of solvent B (97.5% acetonitrile, 0.1% formic acid).

As peptides eluted they were subjected to electrospray ionisation and then entered into an LTQ Velos ion-trap mass spectrometer (Thermo Fisher, San Jose, CA). Peptides were detected, isolated and fragmented to produce a tandem mass spectrum of specific fragment ions for each peptide. Dynamic exclusion was enabled such that ions were excluded from reanalysis for 30 s. Peptide sequences (and hence protein identity) were determined by matching protein databases with the acquired fragmentation pattern by the software program, Sequest (Thermo Fisher, San Jose, CA). The human IPI database (Ver. 3.6) was used for searching. Precursor mass tolerance was set to ± 2.0 Da and MS/MS tolerance was set to 1.0 Da. A reversed-sequence database was used to set the false discovery rate at 1%. Filtering was performed using the Sequest primary score, Xcorr and delta-Corr. Spectral matches were further manually examined and multiple identified peptides ( > = 2) per protein were required.

**SAINT analysis of AP-MS data.** Two biological repeats were performed for each IAV protein complex under IAV and mock infection conditions, respectively. The resulting data are presented in Supplementary Data 1 and 6 and were compared with 77 vector controls (cells transfected with empty vector), our published database of 101 protein complexes[12–14] and 60 unpublished protein complexes that were purified from stable HEK293 cell lines expressing FLAG tag-fused non-viral proteins handled in identical fashion (The whole data set used for SAINT analysis is included in Supplementary Data 3). Proteins found in the control group were considered as non-specific binding proteins. The SAINT algorithm (http://sourceforge.net/projects/saint-apms) was used to evaluate the MS data[11]. The SAINT utilizes label-free quantitative information to compute confidence scores (probability) for putative interactions[11,51,52]. Such quantitative information can include counts (for example, spectral counts or number of unique peptides). In an optimal setting, SAINT utilizes negative control immunoprecipitation data (typically, purifications without expression of the bait protein or with expression of an unrelated protein) to identify non-specific interactions in a semi-supervised manner. A separate unsupervised SAINT modelling is capable of scoring interactions in the absence of implicit control data, but only when a sufficient number of experiments are used for the modelling[52]. The default SAINT options were low Mode = 1, min Fold = 0 and norm = 0. The SAINT scores computed for each biological replicate were averaged (AvgP) and reported as the final SAINT

score. The fold change was calculated for each prey protein as the ratio of spectral counts from replicate bait purifications to the spectral counts of the same prey protein across all negative controls. A background factor of 0.1 was added to the average spectral counts of negative controls to prevent division by zero. The proteins included in the final interactome list had an AvgP ≥ 0.89. The threshold for SAINT scores was selected based on receiver operating curve analysis performed using publicly available protein interaction data. A SAINT score of AvgP ≥ 0.89 was considered a true positive BioID protein with an estimated FDR of ≤ 2%. Proteins with SAINT score < 0.89 are considered as non-specific binding proteins. We manually removed ribosomal proteins from the final HCIP list because these proteins are prone to associate with RNA-binding proteins.

The results reported here will require additional validation using more directed experiments. Low affinity or transient, and may be lost during the stringent washing procedures with lysis buffer. Although spectral counts have a correlation with protein abundance other methods, for example, SILAC offer more precise strategies for quantitative proteomics[53]. Nevertheless, by mapping the IAV–host interactomes using AP-MS, we were able to reveal novel common and strain-specific interactors.

**Statistical analysis.** The sample size was sufficient for data analysis using paired two-tailed Student's $t$-test. For all statistical analysis, differences were considered to be statistically significant at values of $P < 0.05$.

**Bioinformatic analysis.** The Search Tool for the Retrieval of Interacting Genes/Proteins (STRING) database (http://string.embl.de) and curated literature[5–8,17,54,55] were used to identify novel HCIPs. The IAV–host protein interaction network was generated in Cytoscape (www.cytoscape.org). DAVID Bioinformatics Resources 6.8 (ref. 56) was used to perform Gene Ontology enrichment analysis of the 625 HCIPs from comparative IAV–host protein interactomes. Terms within the top 5 'GOTERM_BP_DIRECT' category were used.

**Data availability.** The authors declare that the data supporting the findings of this study are available within the article and its Supplementary Information files, and from the corresponding author upon request. The AP-MS data that support the findings of this study have been deposited in the IMEx (http://www.imexconsortium.org) consortium through IntAct[57] with the accession code IM-25584 and also can be downloaded from https://sharehost.hms.harvard.edu/mbib/dorf/nature_communications.

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

## Acknowledgements

We thank Michael Berman for expert technical assistance, Drs Peter Palese (Mount Sinai School of Medicine, NY, NY), Adolfo Garcia-Sastre (Mount Sinai School of Medicine, NY, NY), Charles Russell (St Jude's Hospital, Memphis, TN), Sean Whelan (Harvard Medical School, Boston, MA), Robert Webster (St Jude Children's Hospital, Memphis, TN), Ervin Fodor (University of Oxford, Oxford, UK), Andrew Rice (Baylor College of Medicine, Houston, TX), Koyu Hara (Kurume University School of Medicine, Japan) and Andrew Kowalczyk (Emory University, Atlanta, GA) for reagents. This work was supported by NIH grant R01AI089829 (M.D.), R01AI121288 (M.D. and S.L.), R01HL116876 (L.L.), R21AI121591 (L.L.), RAC Fund (S.L.) and NIH pilot grant P20GM103648 (S.L.).

## Author contribution

Conceived and designed the experiments: L.W., M.E.D., S.L. Performed the experiments: L.W., B.F., W.L., G.P. Analysed the data: L.W., B.F., S.L. Wrote the paper: L.W., L.L., M.E.D., S.L.

## Additional information

**Competing financial interests:** The authors declare no competing financial interests.

