## [Peer Review File · Nature Communications]

Reviewers' comments:

Reviewer #1 (Remarks to the Author):

The manuscript by Wang et al presents a comparative analysis of protein interactions networks of four different influenza A virus strains in human cells. The authors use immunoprecipitations and proteomics to identify interactomes of viral proteins in infected and uninfected cells and identify protein interactions common and unique to the four different influenza A virus strains. By performing over-expression and RNAi experiments, within the 'core' interactome, consisting of common interactions, they identify six novel host factors that regulate influenza A virus infection. Among these, plakophilin 2 (PKP2) is shown to restrict influenza A virus replication by interfering with the assembly of the viral RNA polymerase complex.

Similar large-scale studies of interactomes of influenza A virus proteins have been performed by a number of groups (for example Watanabe et al Cell Host Microbe 2014). The strength of the study by Wang et al is that it carries out this analysis in a comparative manner, considering four different viral strains. In addition, interactomes both in infected and uninfected cells are considered. Furthermore, it also identifies several factors regulating viral replication, including PKP2 that is convincingly shown to function as a restriction factor. Generally, the results are clearly presented and the results fully support the conclusions. The study will be of interest to the influenza research community and will particularly stimulate further research into the role of the host factors identified.

Specific points:

1. Line 62; "infected with or without 1 MOI of PR8 IAV" should read "mock infected or infected with 1 MOI of PR8 IAV".
2. Fig. 1b,c; Fig. S2b-d; what do the empty and full bars indicate?
3. Fig. 2b; blue and red colours need to be specified. The time of infection is not given and it is unclear how Z scores were calculated (lines 525-526); what is the relevance of the authors' database? what are the "205 selective controls"? This should be explained.
4. Fig. 2c; for every figure presenting statistics it should be stated how many biological and technical repeats the authors are considering (including Fig. 4a,b,e,f; Fig. 5a-d; Fig S3,b,c; Fig. S4a,b).
5. Line 174; "Fig. 3d" should read "Fig. 3c".
6. Fig. 3e; this figure shows that both PB1 and PKP2 are present in the nucleus but this does not indicate that PKP2 interacts with PB1 in the nucleus (lines 141-143).
7. Fig. 6d; the authors do not specify whether the IAV complex is derived from infections or from transfections (lines 179-180).
8. Fig. 5a; how was mRNA measured? presumably by RT-PCR but it is not stated (lines 559-560).

Reviewer #2 (Remarks to the Author):

Host proteins play important roles in influenza A virus (IAV) replication in host cells - some functioning to promote virus replication and others acting as part of the cellular defense against infection - and interactions between viral and cellular proteins are critical for these processes. Multiple published studies describe IAV-host interactomes for individual IAV strains. However, systematic identification of interactome similarities and differences across different IAV strains is currently lacking. Strain-specific interactome differences may shed light on mechanisms of differential pathogenicity, while interactome characteristics that are conserved across many strains could serve as potential targets for broad spectrum therapeutics aimed at affecting host processes.

In this manuscript, the authors report an affinity purification mass spectrometry (AP-MS) approach to identify IAV-host interactions for 11 proteins from the PR8 (H1N1) laboratory strain, 5 proteins from two additional H1N1 strains (the WSN laboratory strain and an isolate from the 2009 pandemic) and 2 proteins from an historic H3N2 strain isolated in 1968. They discovered strain-specific and common interactions between viral and cellular proteins, some of which are induced by viral infection. A subset of these interactions have not been identified in any previous study.

Subsequent analyses focused on a "core" set of 195 IAV-host interactions that were observed for all 4 virus strains (referred to as 'HCIPs'). Of these, 37 HCIPs were selected for interaction validation by co-immunoprecipitation (98% [36 of 37] of the interactions were validated) and for screening for effects of overexpression on IAV replication. Overexpression of one factor (EIF2B4) resulted in increased virus replication (suggesting pro-viral activity), and overexpression of 5 others (FKBP8, PKP2, STUB1, TRIM41 and ZMPSTE24) resulted in decreased virus replication (suggesting antiviral activity). Knockdown of each of these factors using RNA interference (RNAi) resulted in the opposite effect of overexpression, confirming their presumed pro- and anti-viral activities.

The authors then focused on a PB1 interaction partner (PKP2) - i.e., plakophilin 2, a protein that may regulate cellular adhesion processes and beta-catenin signaling - for in-depth analysis (PB1 is one of the subunits of the IAV polymerase complex). They confirmed that endogenous PKP2 interacts with PB1 in IAV-infected primary tracheal epithelial cells, and that PKP2 co-immunoprecipitates with all four PB1 proteins used for the initial AP-MS analysis. Detailed RNAi and overexpression studies demonstrated that PKP2 clearly impacts IAV replication by acting in an antiviral manner. Interaction domains were mapped to the N-terminus of PKP2 and the C-terminus of the PB1 protein (which is required for interaction with PB2, another component of the IAV polymerase complex); and simultaneous overexpression of PKP2, PB1 and PB2 resulted in reduced interaction between PB1 and PB2, leading the authors to hypothesize that PKP2 competes with PB2 for interaction with PB1 to impair viral polymerase activity. Consistent with this hypothesis, PKP2 overexpression impairs IAV gene expression in a mini-replicon system; however, while PKP2 probably associates with the viral polymerase complex in infected cells, overexpression does not appear to result in appreciable dissolution of PB1-PB2 complexes. Here, it should be noted that the authors concluded that PB1-PB2 complexes are appreciably altered in infected cells overexpressing PKP2, but this reviewer does not agree with their conclusion.

Overall, this manuscript attempts to address an interesting and important question in influenza virology; most of the experiments appear to be well-designed; and the associated data might be useful for clarifying IAV-host protein interaction networks that regulate pathogenicity and/or for identifying targets for development of novel countermeasures. However, there are several facets of this study that reduce enthusiasm for publication in its current form:

- ♣ The IAV strains that were examined are not ideal for elucidating strain-dependent similarities and differences of IAV-host interactomes in humans. Two (PR8 and WSN) are laboratory strains with complicated passage histories and tropism/pathogenicity profiles in animal models that may not reflect that of recent authentic human isolates; three viruses are of the H1N1 subtype, which limits the ability to perform cross-subtype comparisons; and the H3N2 strain is not a recent isolate.

- ♣ Little information on differences in pathogenicity or replicative abilities of these viruses is provided, so it is difficult to see how the interactome comparison will inform understanding of virus-host interactions for different influenza strains/subtypes.

- ♣ The lack of full interactome datasets for all 4 viruses used in this study further limits the ability to perform comprehensive and systematic comparisons across different virus subtypes or strains.

- ♣ The overall goal of this study was to explore comparative virus/host protein interaction networks, but the actual comparative analysis that was performed was minimal. This is a missed opportunity. Moreover, while the authors conclude that "comparative analyses of the influenza-host protein interactomes identified PKP2 as a natural inhibitor of IAV polymerase complex," it is important to note that the PKP2-PB1 interaction - and the effect of PKP2 overexpression on IAV replication - had already been established in a study published previously by the authors (PMID: 26057645). Thus, the comparative analysis described here was not essential for identifying PKP2's antiviral activity against IAV other than demonstrating interaction with PB1 proteins from multiple influenza virus strains.

- ♣ To further strengthen the comparison of different IAV/host interactome networks, overexpression and RNAi studies should be extended to additional virus strains, rather than focusing only on effects in the context of PR8/WSN infections.

- ♣ The presented mechanistic studies do not convincingly support the conclusions drawn by the authors, and additional studies are needed to fully establish how PKP2 exerts its antiviral effects against IAV. In particular, PKP2 may be involved in regulating beta-catenin signaling, which in turn can affect antiviral signaling pathways. The possibility of a link between PKP2 and beta-catenin in IAV-infected cells should be examined.

For the reasons described above, this manuscript is more suitable for virology journals. Additional specific comments are included below.

SPECIFIC COMMENTS

1. Lines 43-47; "Analysis of the network revealed that NS1, M2, PB1, PB2 and NP are the major nodes connecting cellular factors with known and predicted roles in immunity and viral infection. Thus, we further mapped the protein interactomes of these viral proteins from two other H1N1 strains (WSN/33, A/WSN/1933 (H1N1); NY/2009, A/New York/1682/2009 (H1N1)) and one H3N2 strain..."

a. This statement is not accurate, since only NS1 and NP were analyzed for the H3N2 strain.

b. Also, NS1 appears to be no more major than HA in the PPI network shown in Figure S1b. Why wasn't HA from all virus strains examined?

2. Lines 49-50; The authors may want to consider revising the phrase "functional analysis" since this phrase is frequently used to describe pathway or process enrichment analysis of 'omics datasets.

3. Lines 55-56; "Our study demonstrates that PKP2 disrupts the interaction between influenza PB1 and PB2, thereby disintegrating IAV polymerase complex and limiting viral replication." Data from infected cells (Figure 6d) does not support this conclusion (see also point 17 below).

4. Line 62; 'MOI' needs to be spelled out.

5. For cells stably expressing viral proteins, used for AP-MS studies

a. Are there virus protein-induced effects on cell viability?

b. How does stable overexpression of the viral protein affect virus replication? This is important to consider, since alterations in the replication cycle could affect the composition of the interaction dataset.

6. Line 64; "The AP-MS of each IAV protein was biologically repeated once." Does this mean there were two biological replicates? If so, did the authors keep only PPIs that were identified in both replicates? See also point 19-I below.

7. Lines 64-66;

a. The authors should consider including a very brief description of how the SAINT statistical analysis approach works.

b. "...large in-house proteomic database derived from HEK293 cells (161 protein complexes)." What does this mean?

8. Figure S1b and 2a; The dotted lines indicate interactions that were induced by IAV

infection, as indicated by the legend. Please indicate in the legend what solid lines represent. Specifically, do they represent interactions that occurred in both infected and uninfected cells?

9. Figure S2a; The authors should show expression levels of IAV proteins in all stable cell lines used for the study.

10. Figure 1b-c and Figure S2b-d; The authors use NSAF scores to compare the relative abundance of individual host factors in affinity purified preparations of viral proteins from different influenza strains, and use this score as a proxy for estimating strain-specific similarities and differences in virus-host protein interactions. Were validation experiments performed to demonstrate that NSAF scores accurately reflect differences in the level of interaction between virus strains? Were experiments performed to determine whether viral proteins from different strains were similarly affinity purified?

11. Figure S3; It's not clear how the pathway analysis was done. Was the core set of 195 proteins used as a group for analysis? What was the criteria for inclusion of enrichments in Fig. S3a?

12. Lines 114-116; "Furthermore, co-immunoprecipitation verified 48 out of 49 (98%) interactions between 37 HCIPs and viral proteins (Supplementary Table 3), indicating the high quality of the core interactome." Performing confirmation co-immunoprecipitations for 37 HCIPs with viral proteins seems arbitrary. How were these 37 factors selected for validation? Were the confirmation immunoprecipitations performed in stable PB1-expressing cells infected with IAV, or were interaction validations carried out with overexpressed PB1 and HCIPs?

13. Lines 117-119 and Fig. 2b; Did the authors check expression levels of overexpressed HCIPs in this experiment? Did overexpression affect cell viability? What was the HCIP transfection efficiency?

14. Lines 125-126; "RNAi knockdown efficiency and cell viability were verified (Supplementary Fig. 3c)." This is data for A549 cells. The authors also need to show viability data for primary tracheal cells.

15. Lines 177-179; "We also noted that PA failed to pull down PKP2, suggesting the interaction specificity between PB1 and PKP2 (Fig. 6c)." However, NP did pull down PKP2 (Fig. 6b), suggesting lack of specificity. Further, interaction between PKP2 and NP was not observed in the original screen. Can the authors explain this observation? Were control (IgG) immunoprecipitations performed for these experiments?

16. Line 179 and Figure 6d; "To investigate the effect of PKP2 on the IAV polymerase complex..." Figure 6d would be improved by showing blots of NP and PA.

17. Lines 182-184; "Furthermore, PKP2 perturbed the fraction distribution of IAV polymerase complex and reduced the PB2/PB1 protein ratio in fractions containing PKP2

(Fig. 6d)." PB1 and PB2 distribution look about the same between PKP2 and GFP cells and the reported differences in PB2/PB1 ratio are moderate at best. This data does not support the hypothesis that PKP2 disrupts the interaction between PB2 and PB1 infected cells. How many times was this experiment performed, and are the results reproducible?

18. Lines 195-196; "Notably, NS1 of NY/2009 weakly interacts with CPSF even though it possesses intact F103 and M106, suggesting that other binding sites are also required." There could be many other reasons for reduced CPSF interaction (e.g., technical differences between experiments with IAV proteins from different virus strains or interactions that are differentially affected by other virus proteins).

19. Methods

a. The authors need to indicate the dilutions at which antibodies were used.

b. The authors need to identify the source of the HCIPs (plasmids) other than PKP2 (many were used to produce the data shown in Figure 2b-c). They also need to identify the source of mutant PB1 plasmids, H3N2 influenza gene plasmids, the control plasmid used for overexpression studies, and the plasmid expressing CTNNB1.

c. Line 281; 'TAP' needs to be spelled out.

d. What was the amount of HCIP plasmid DNA used in transfections for overexpression experiments?

e. Three siRNAs against PKP2 are shown in Figure 5a, but only a single PKP2 siRNA is described in the Methods.

f. The scrambled control siRNA is not described.

g. The amount of siRNA used for transfections in each cell type needs to be described.

h. Line 340 indicates that MEFs were used, but the rest of the manuscript describes primary tracheal cells, which are not mentioned. This needs to be reconciled.

i. Lines 352-358; Amounts of plasmids used for the polymerase reconstitution assay need to be described. Also, the use of the Renilla luciferase control needs to be explained (e.g., did they use Renilla data to normalize for transfection efficiency?).

j. For AP-MS experiments, at what time post-infection were lysates collected?

k. Lines 363-364; "Each group of cells was expanded and cultured in five 15-cm dishes." The significance of using 5 dishes here is not clear.

l. Line 375; "Two independent purifications of each tagged viral protein complex were analyzed by AP-MS." Were these independent purifications performed at the same time (technical replicates) or at different times (biological replicates)? If biological replicates were

performed, were control AP-MS experiments (without infection) performed in duplicate, as well?

m. Lines 376-377; "...our database of 77 controls and 161 samples..." Does this mean there are 161 samples from cells overexpressing 77 different proteins? Note that line 66 refers to "...(161 protein complexes)..."

n. Lines 378-379; "Proteins found in the control group were considered as non-specific binding proteins." To clarify, non-specific binding proteins were those that were identified only in control (transfected) cells, correct?

o. Details of mass spectrometry data generation and processing need to be described.

p. Lines 382-383; "The fold change was calculated for each prey protein as the ratio of spectral counts from replicate bait purifications to the spectral counts across all negative controls." Does this mean all negative controls for an individual protein or all negative controls for all proteins? If a protein was identified in only one replicate, was it maintained in the resulting network?

q. Line 387; "...the FLAG AP-MS data set as a list of true positive interactions..." it's not clear what the authors are referring to here. Is this the "database of 77 controls and 161 samples"? If so, what evidence indicates that these are "true positive interactions"?

r. Lines 392-396; To clarify, were STRING, InACT and BioGRID databases used to determine whether host proteins are known to interact with influenza virus proteins? If so, do these databases include data from the following publications? PMID: 25464832, PMID: 21715506, PMID: 26651948 (among others)

s. The authors need to describe how they performed functional enrichment analysis.

t. The authors need to describe the generation of stable cell lines.

20. Figure 3b would be better if it showed interaction between endogenous PKP2 and influenza PB1 proteins in infected cells, similar to the experiment shown in Figure 3a.

21. Figure Legends

a. Line 524-525; How long after infection were luciferase assays performed?

b. Line 526; "... our laboratory database containing results from 205 selective controls..." What are 'selective' controls?

c. For all data, the authors should indicate how many independent replicate experiments were performed in the figure legends and whether the data in the figures shows data combined across experiments.

- d. Figure 4d and 5c; How many cells were counted for each condition?
- e. Figure 4f; How long after plasmid transfection were infections performed?
- f. Figure 6a-c; At what time after transfection were lysates harvested?

Reviewer #3 (Remarks to the Author):

I was asked to comment specifically on the proteomics part of the manuscript.

The authors describe the study of the host proteins interacting with 11 proteins from one strain of the Influenza A virus, and follow up with the analysis of the interactomes of five of these proteins from two other Influenza A strains. They identify 6 novel host factors regulating viral infection and then focus their detailed analysis on plakophilin 2, demonstrating its involvement in the defense against the virus. The authors use AP-MS, which is a great methodology to determine the interactomes. The findings are interesting and biologically relevant. However, there are several points, which should be clarified before the manuscript is accepted for publication.

Major concerns:

1. It is unclear from the description in the Methods section and from Figure S1a (experimental workflow) whether the analyses were performed only once, or whether there were replicates. The Methods section (page 17) states: "Two independent purifications of each tagged viral protein complex were analyzed by AP-MS" - does it mean, one analysis for the cells that were subsequently infected, and one for uninfected (Figure S1a would suggest so)? Or were there replicates of each of the infected and uninfected conditions? Replicates are obviously recommended for each tagged viral protein. Otherwise it is impossible to determine the variability between individual experiments and the results may be attributed to this variability.
2. Spectral count is probably the least reliable method of quantification. Given that the authors used cultured cells, it should have been easy to label the cells metabolically, or, if this was impossible, use another method of label-free quantification.

Minor issues:

1. Please check the spelling and grammar throughout.
2. Page 3, line 43: I would refrain from using the word "genes" in the context of interactome, because the proteins, and not genes, are the players here.
3. Page 7, line 141 - a citation is missing (consistent with a previous report - which report?)

Response to the review comments

We appreciate the helpful comments of the reviewers. We have revised the manuscript to address each of the listed concerns. We have also performed the experiments that the reviewers requested. New figures (Fig. 6c, 6d, Fig. S1b-S1d, Fig. S3a-S3e, Fig. S4a-S4c, Fig. S4d, Fig. S6a-S6f, Fig. S7a-S7b, Fig. S8, and Fig. S10a-S10b) have been added to the revision. Additional figures and tables (Fig. 1a-1c, Fig. 2a-2b, Fig. S2 and Fig. S5a-S5c and Supplementary Tables 1-3) have been modified. Point-by-point reply is highlighted in red.

Reviewer #1:

The manuscript by Wang et al presents a comparative analysis of protein interactions networks of four different influenza A virus strains in human cells. The authors use immunoprecipitations and proteomics to identify interactomes of viral proteins in infected and uninfected cells and identify protein interactions common and unique to the four different influenza A virus strains. By performing over-expression and RNAi experiments, within the 'core' interactome, consisting of common interactions, they identify six novel host factors that regulate influenza A virus infection. Among these, plakophilin 2 (PKP2) is shown to restrict influenza A virus replication by interfering with the assembly of the viral RNA polymerase complex.

Similar large-scale studies of interactomes of influenza A virus proteins have been performed by a number of groups (for example Watanabe et al Cell Host Microbe 2014). The strength of the study by Wang et al is that it carries out this analysis in a comparative manner, considering four different viral strains. In addition, interactomes both in infected and uninfected cells are considered. Furthermore, it also identifies several factors regulating viral replication, including PKP2 that is convincingly shown to function as a restriction factor. Generally, the results are clearly presented and the results fully support the conclusions. The study will be of interest to the influenza research community and will particularly stimulate further research into the role of the host factors identified.

We thank the reviewer's positive comments and helpful suggestions. We addressed the concerns point-by-point as below.

Specific points:

1. Line 62; "infected with or without 1 MOI of PR8 IAV" should read "mock infected or infected with 1 MOI of PR8 IAV".

Corrected (line 67-68).

2. Fig. 1b,c; Fig. S2b-d; what do the empty and full bars indicate?

There are the same bars. We now make it clearer by changing the gradient fill to solid fill. (Fig. 1b, 1c, S5a-S5c).

3. Fig. 2b; blue and red colours need to be specified. The time of infection is not given and it is unclear how Z scores were calculated (lines 525-526); what is the relevance of the authors' database? what are the "205 selective controls"? This should be explained.

We now use one color for Fig.2b and add the time of infection (16 hr, line 664). Because there are some unpublished host factors in the 205 controls, we now use 34 host factors from IAV-host interactomes, 7 empty vectors and 26 genes from our previous innate immunity protein interactome (PMID: 21903422). All these gene names are listed in Supplementary Table 3. Due to the relative small size of dataset, we now adopt standard deviation to determine the outstanding hits ($\leq 2xSD$ or $\geq 2xSD$ from mean) instead of Z scores.

4. Fig. 2c; for every figure presenting statistics it should be stated how many biological and technical repeats the authors are considering (including Fig. 4a,b,e,f; Fig. 5a-d; Fig S3,b,c; Fig. S4a,b).

Biological repeats and statistics method now are stated in these Figures.

5. Line 174; "Fig. 3d" should read "Fig. 3c".

We thank the reviewer for critical reading. Now its is corrected (line 180).

6. Fig. 3e; this figure shows that both PB1 and PKP2 are present in the nucleus but this does not indicate that PKP2 interacts with PB1 in the nucleus (lines 141-143).

We agree with the reviewer and now change the statement to co-localization in nucleus (line 144-145).

7. Fig. 6d; the authors do not specify whether the IAV complex is derived from infections or from transfections (lines 179-180).

We now specify that the IAV complex is derived from infections (line 185-187).

8. Fig. 5a; how was mRNA measured? presumably by RT-PCR but it is not stated (lines 559

560).

mRNA was measured by RT-PCR. Now it is stated (Line 708)

Reviewer #2:

Host proteins play important roles in influenza A virus (IAV) replication in host cells - some functioning to promote virus replication and others acting as part of the cellular defense against infection - and interactions between viral and cellular proteins are critical for these processes. Multiple published studies describe IAV-host interactomes for individual IAV strains. However, systematic identification of interactome similarities and differences across different IAV strains is currently lacking. Strain-specific interactome differences may shed light on mechanisms of differential pathogenicity, while interactome characteristics that are conserved across many strains could serve as potential targets for broad spectrum therapeutics aimed at affecting host processes.

In this manuscript, the authors report an affinity purification mass spectrometry (AP-MS) approach to identify IAV-host interactions for 11 proteins from the PR8 (H1N1) laboratory strain, 5 proteins from two additional H1N1 strains (the WSN laboratory strain and an isolate from the 2009 pandemic) and 2 proteins from an historic H3N2 strain isolated in 1968. They discovered strain-specific and common interactions between viral and cellular proteins, some of which are induced by viral infection. A subset of these interactions have not been identified in any previous study.

Subsequent analyses focused on a "core" set of 195 IAV-host interactions that were observed for all 4 virus strains (referred to as 'HCIPs'). Of these, 37 HCIPs were selected for interaction validation by co-immunoprecipitation (98% [36 of 37] of the interactions were validated) and for screening for effects of overexpression on IAV replication. Overexpression of one factor (EIF2B4) resulted in increased virus replication (suggesting pro-viral activity), and overexpression of 5 others (FKBP8, PKP2, STUB1, TRIM41 and ZMPSTE24) resulted in decreased virus replication (suggesting antiviral activity). Knockdown of each of these factors using RNA interference (RNAi) resulted in the opposite effect of overexpression, confirming their presumed pro- and anti-viral activities.

The authors then focused on a PB1 interaction partner (PKP2) - i.e., plakophilin 2, a protein that may regulate cellular adhesion processes and beta-catenin signaling - for in-depth analysis (PB1 is one of the subunits of the IAV polymerase complex). They confirmed that endogenous PKP2 interacts with PB1 in IAV-infected primary tracheal epithelial cells, and that PKP2 co-immunoprecipitates with all four PB1 proteins used for the initial AP-MS analysis. Detailed RNAi and overexpression studies demonstrated that PKP2 clearly impacts IAV replication by acting in an antiviral manner. Interaction domains were mapped to the N-terminus of PKP2 and the C-terminus of the PB1 protein (which is required for interaction with PB2, another component of the IAV polymerase complex); and simultaneous overexpression of PKP2, PB1 and PB2 resulted in reduced interaction between PB1 and PB2, leading the authors to hypothesize that PKP2 competes with PB2 for interaction with PB1 to impair viral polymerase activity. Consistent with this hypothesis, PKP2 overexpression impairs IAV gene expression in a mini-replicon system; however, while PKP2 probably associates with the viral polymerase complex in infected cells, overexpression does not appear to result in appreciable dissolution of PB1-PB2 complexes. Here, it should be noted that the authors concluded that PB1-PB2 complexes are appreciably altered in infected cells overexpressing PKP2, but this reviewer does not agree with their conclusion.

We repeated the fractionation experiment and performed co-IP to demonstrate the specific interaction between PB1 and PKP2. The new data is now shown in Fig.6d and Fig. S10a. Although we believe that the new data support PKP2 perturbation of IAV polymerase complex, we now emphasize that PKP2 inhibits IAV polymerase activity (line 59).

Overall, this manuscript attempts to address an interesting and important question in influenza virology; most of the experiments appear to be well-designed; and the associated data might be useful for clarifying IAV-host protein interaction networks that regulate pathogenicity and/or for identifying targets for development of novel countermeasures. However, there are several facets of this study that reduce enthusiasm for publication in its current form:

The IAV strains that were examined are not ideal for elucidating strain-dependent similarities and differences of IAV-host interactomes in humans. Two (PR8 and WSN) are laboratory strains with complicated passage histories and tropism/pathogenicity profiles in animal models that may not reflect that of recent authentic human isolates; three viruses are of the H1N1 subtype, which limits the ability to perform cross-subtype comparisons; and the H3N2 strain is not a recent isolate.

We thank the reviewer for the constructive suggestion. We now add the protein interactome of a recent authentic human isolate, A/Viet Nam/1203/2004 (H5N1). This virus is classified into biosafety level 3 (BSL-3) and thus infection with this strain cannot be performed in my and my collaborators' labs (all are BSL-2). Nonetheless, cross-subtype comparisons of the protein interactomes are performed as shown in Fig.1, Fig. S5 and Supplementary Table 2.

Little information on differences in pathogenicity or replicative abilities of these viruses is provided, so it is difficult to see how the interactome comparison will inform understanding of virus-host interactions for different influenza strains/subtypes.

We now present one pandemic strain (A/New York/1682/2009) and one emerging highly pathogenic strain (A/Viet Nam/1203/2004). The pathogenicity of these strains have been reported by Dr. Tumpey's lab (PMID: 21047961). We add the information with two references in the Introduction (line 49-50).

The lack of full interactome datasets for all 4 viruses used in this study further limits the ability to perform comprehensive and systematic comparisons across different virus subtypes or strains.

It took more than 4 years to complete the current interactomes.

The overall goal of this study was to explore comparative virus/host protein interaction networks, but the actual comparative analysis that was performed was minimal. This is a missed opportunity. Moreover, while the authors conclude that "comparative analyses of the influenza-host protein interactomes identified PKP2 as a natural inhibitor of IAV polymerase complex," it is important to note that the PKP2-PB1 interaction - and the effect of PKP2 overexpression on IAV replication - had already been established in a study published previously by the authors (PMID: 26057645). Thus, the comparative analysis described here was not essential for identifying PKP2's antiviral activity against IAV other than demonstrating interaction with PB1 proteins from multiple influenza virus strains.

PKP2 has not been reported as an antiviral protein or a PB1 interactor previously either by us or others. Our paper (PMID: 26057645) reported that TRIM32 is an E3 ligase for PB1. We are sorry for the confusion.

To further strengthen the comparison of different IAV/host interactome networks, overexpression and RNAi studies should be extended to additional virus strains, rather than focusing only on effects in the context of PR8/WSN infections.

We now repeated the experiments with the pandemic IAV and the new data are shown in the revised Fig. S7a.

The presented mechanistic studies do not convincingly support the conclusions drawn by the authors, and additional studies are needed to fully establish how PKP2 exerts its antiviral effects against IAV. In particular, PKP2 may be involved in regulating beta-catenin signaling, which in turn can affect antiviral signaling pathways. The possibility of a link between PKP2 and beta-catenin in IAV-infected cells should be examined.

We thank the reviewer for another helpful suggestion. We examined the effects of PKP2 alone or in combination with beta-catenin on the Wnt signaling pathway and found that PKP2 had minimal effects on the Wnt signaling. The data is presented in Fig. S8.

For the reasons described above, this manuscript is more suitable for virology journals. Additional specific comments are included below.

SPECIFIC COMMENTS

1. Lines 43-47; "Analysis of the network revealed that NS1, M2, PB1, PB2 and NP are the major nodes connecting cellular factors with known and predicted roles in immunity and viral infection. Thus, we further mapped the protein interactomes of these viral proteins from two other H1N1 strains (WSN/33, A/WSN/1933 (H1N1); NY/2009, A/New York/1682/2009 (H1N1)) and one H3N2 strain..."

a. This statement is not accurate, since only NS1 and NP were analyzed for the H3N2 strain. We now change the statement to make it accurate (line 48-49).

b. Also, NS1 appears to be no more major than HA in the PPI network shown in Figure S1b. Why wasn't HA from all virus strains examined?

NS1 is a known multifunctional viral protein. We now add this to the reason why NS1 is chosen, but not HA (line 46-47).

2. Lines 49-50; The authors may want to consider revising the phrase "functional analysis" since this phrase is frequently used to describe pathway or process enrichment analysis of 'omics datasets.

We change it to "gain- and loss-of-function studies" (line 52).

3. Lines 55-56; "Our study demonstrates that PKP2 disrupts the interaction between influenza PB1 and PB2, thereby disintegrating IAV polymerase complex and limiting viral replication." Data from infected cells (Figure 6d) does not support this conclusion (see also point 17 below).

We repeated the experiment and the results are reproducible. We now show the new data in Fig. 6d. The data clearly show there are 2 to 3-fold differences. Although we believe that the new data support our conclusion, we soften the tone to state that PKP2 alters the integrity of IAV polymerase complex (line 194).

4. Line 62; 'MOI' needs to be spelled out.

Corrected (line 68).

5. For cells stably expressing viral proteins, used for AP-MS studies

a. Are there virus protein-induced effects on cell viability?

The viral proteins have minimal effects on cell viability as shown in the new Figures, S1c and S4a.

b. How does stable overexpression of the viral protein affect virus replication? This is important to consider, since alterations in the replication cycle could affect the composition of the interaction dataset.

Viral replication in stable cell lines is comparable to that in the control cell line transfected with empty vector. The new data are shown in Fig. S1d and S4b.

6. Line 64; "The AP-MS of each IAV protein was biologically repeated once." Does this mean there were two biological replicates? If so, did the authors keep only PPIs that were identified in both replicates? See also point 19-I below.

There are two biological replicates of each tagged viral protein under each condition, the infected and uninfected, respectively. We now clarify it in Results and Methods ((line 70-71; 454-455). All HICPs were identified in both replicates.

7. Lines 64-66;

a. The authors should consider including a very brief description of how the SAINT statistical analysis approach works.

Added (line 461-468).

b. "...large in-house proteomic database derived from HEK293 cells (161 protein complexes)." What does this mean?

We compared the IAV proteomic data with our proteomic database including 77 vector controls (cells transfected with empty vector), our published database of 101 protein complexes (PMID: 21903422, PMID: 24778252, PMID: 24060851) and 60 unpublished protein complexes. (line 72-73, line 456-459, the AP-MS dataset including 77 controls and 161 samples now is presented in Supplementary Table 1, sheet e).

8. Figure S1b and 2a; The dotted lines indicate interactions that were induced by IAV infection, as indicated by the legend. Please indicate in the legend what solid lines represent. Specifically, do they represent interactions that occurred in both infected and uninfected cells?

The straight solid lines represent interactions that occurred in both infected and uninfected cells. We now add it in the Figure legend.

9. Figure S2a; The authors should show expression levels of IAV proteins in all stable cell lines used for the study.

We now include the IAV protein expression levels in all cell lines (Fig. S1b and Fig. S3a-3e).

10. Figure 1b-c and Figure S2b-d; The authors use NSAF scores to compare the relative abundance of individual host factors in affinity purified preparations of viral proteins from different influenza strains, and use this score as a proxy for estimating strain-specific similarities and differences in virus-host protein interactions. Were validation experiments performed to demonstrate that NSAF scores accurately reflect differences in the level of interaction between virus strains? Were experiments performed to determine whether viral proteins from different strains were similarly affinity purified?

We validated the differential interactions between CPSF4 and 5 NS1 proteins using stable cell lines. Our data confirm the NSAF score correlates with the level of interaction between virus strains (Fig. S4d). We also show that the NS1 proteins from different strains were similarly affinity purified (Fig. S4d).

11. Figure S3; It's not clear how the pathway analysis was done. Was the core set of 195 proteins used as a group for analysis? What was the criteria for inclusion of enrichments in Fig. S3a?

DAVID Bioinformatics Resources 6.8 was used to perform Gene Ontology enrichment analysis. All 625 HCIPs from comparative IAV-host protein interactomes were used as a group for analysis (Fig. S4c, line 497-500).

12. Lines 114-116; "Furthermore, co-immunoprecipitation verified 48 out of 49 (98%) interactions between 37 HCIPs and viral proteins (Supplementary Table 3), indicating the high quality of the core interactome." Performing confirmation co-immunoprecipitations for 37 HCIPs with viral proteins seems arbitrary. How were these 37 factors selected for validation? Were the confirmation immunoprecipitations performed in stable PB1-expressing cells infected with IAV, or were interaction validations carried out with overexpressed PB1 and HCIPs?

We chose HCIPs that were available from Harvard PlasmID Database. Only some of these HCIPs were PB1 interactors and we validated the interactions by co-expression of IAV genes and HCIP genes in HEK293 cells. We take out the data to avoid the confusion and add new co-IP data to validate the interactions between 5 functional hits (EIF2B4, FKBP8, PKP2, TRIM41 and ZMPSTE24) and the corresponding IAV proteins (Fig. S6b-S6f).

13. Lines 117-119 and Fig. 2b; Did the authors check expression levels of overexpressed HCIPs in this experiment? Did overexpression affect cell viability? What was the HCIP transfection efficiency?

We now show the protein expression levels of the 5 HCIPs that regulate IAV infection and the effects of these HCIPs on cell viability (Fig. S6a).

14. Lines 125-126; "RNAi knockdown efficiency and cell viability were verified (Supplementary Fig. 3c)." This is data for A549 cells. The authors also need to show viability data for primary tracheal cells.

As suggested new experiments showed that these siRNA duplexes had little effects on the viability of primary tracheal cells (Fig. S7b).

15. Lines 177-179; "We also noted that PA failed to pull down PKP2, suggesting the interaction specificity between PB1 and PKP2 (Fig. 6c)." However, NP did pull down PKP2 (Fig. 6b), suggesting lack of specificity. Further, interaction between PKP2 and NP was not observed in the original screen. Can the authors explain this observation? Were control (IgG) immunoprecipitations performed for these experiments?

Fig.6b only examines PB1 interaction with NP and PKP2, but not the interaction between NP and PKP2. We now provided new data (Fig. S10a) that clearly show the specific interaction between PKP2 and PB1. All immunoprecipitations are co-IP.

16. Line 179 and Figure 6d; "To investigate the effect of PKP2 on the IAV polymerase complex..." Figure 6d would be improved by showing blots of NP and PA.

We now show the blots of NP and PA (Fig. 6d).

17. Lines 182-184; "Furthermore, PKP2 perturbed the fraction distribution of IAV polymerase complex and reduced the PB2/PB1 protein ratio in fractions containing PKP2 (Fig. 6d)." PB1 and PB2 distribution look about the same between PKP2 and GFP cells and the reported differences in PB2/PB1 ratio are moderate at best. This data does not support the hypothesis that PKP2 disrupts the interaction between PB2 and PB1 infected cells. How many times was this experiment performed, and are the results reproducible?

We repeated the experiment and the results are reproducible. We now show the new data in Fig. 6d. The data clearly show there are 2 to 3-fold differences (Fig. 6d). Although we disagree that the differences in PB2/PB1 ration are moderate, we soften the tone to state that PKP2 alters the integrity of IAV polymerase complex (line 194).

18. Lines 195-196; "Notably, NS1 of NY/2009 weakly interacts with CPSF even though it possesses intact F103 and M106, suggesting that other binding sites are also required." There could be many other reasons for reduced CPSF interaction (e.g., technical differences between experiments with IAV proteins from different virus strains or interactions that are differentially affected by other virus proteins).

We validated the differential interactions between CPSF4 and various NS1 proteins using stable cell lines (Fig. S4d). Dr. Garica-Sastre's group also reported the weak interaction between NS1 of the 2009 IAV and CPSF4 (PMID: 20444891).

19. Methods

a. The authors need to indicate the dilutions at which antibodies were used.

Added (line 251-264).

b. The authors need to identify the source of the HCIPs (plasmids) other than PKP2 (many were used to produce the data shown in Figure 2b-c). They also need to identify the source of mutant

PB1 plasmids, H3N2 influenza gene plasmids, the control plasmid used for overexpression studies, and the plasmid expressing CTNNB1.

Added (line 266-285 and Supplementary Table 3).

c. Line 281; 'TAP' needs to be spelled out.

It is spelled out in the revision (line 316).

d. What was the amount of HCIP plasmid DNA used in transfections for overexpression experiments?

Added (line 374).

e. Three siRNAs against PKP2 are shown in Figure 5a, but only a single PKP2 siRNA is described in the Methods.

Added (line 377-379).

f. The scrambled control siRNA is not described.

Added (line 381-382).

g. The amount of siRNA used for transfections in each cell type needs to be described.

Added (line 382-383).

h. Line 340 indicates that MEFs were used, but the rest of the manuscript describes primary tracheal cells, which are not mentioned. This needs to be reconciled.

We thank the reviewer for critical reading. Now it is corrected (line 292-294).

i. Lines 352-358; Amounts of plasmids used for the polymerase reconstitution assay need to be described. Also, the use of the Renilla luciferase control needs to be explained (e.g., did they use Renilla data to normalize for transfection efficiency?).

The amounts of plasmids are now added (line 395-398). We examined the viral protein expression to determine the transfection efficiency (Fig. S10b). We eliminated the reference to Renilla.

j. For AP-MS experiments, at what time post-infection were lysates collected?

16 hr. Now it is added in the Methods (line 415).

k. Lines 363-364; "Each group of cells was expanded and cultured in five 15-cm dishes." The significance of using 5 dishes here is not clear.

We now change it to cell number (10^8) (line 416).

l. Line 375; "Two independent purifications of each tagged viral protein complex were analyzed by AP-MS." Were these independent purifications performed at the same time (technical replicates) or at different times (biological replicates)? If biological replicates were performed, were control AP-MS experiments (without infection) performed in duplicate, as well?

There are 2 biological replicates of each tagged viral protein under each condition, the infected and uninfected, respectively. We now clarify it in Results and Methods (line 70-71; 454-455). All AP-MS experiments (infection and without infection) were performed in duplicate.

m. Lines 376-377; "...our database of 77 controls and 161 samples..." Does this mean there are 161 samples from cells overexpressing 77 different proteins? Note that line 66 refers to "... (161 protein complexes)..."

We now change it to "were compared with 77 vector controls (cells transfected with empty vector), our published database of 101 protein complexes (PMID: 21903422, PMID: 24778252, PMID: 24060851) and 60 unpublished protein complexes that were purified from stable HEK293 cell lines expressing FLAG tag-fused non-viral proteins handled in identical fashion" (line 456-459, the AP-MS dataset including 77 controls and 161 samples now is presented in Supplementary Table 1, sheet e).

n. Lines 378-379; "Proteins found in the control group were considered as non-specific binding proteins." To clarify, non-specific binding proteins were those that were identified only in control (transfected) cells, correct?

Proteins found in control (cells transfected with empty vector) are all non-specific binding proteins. Proteins in viral protein complexes with a SAINT score of <0.89 are also considered as non-specific binding proteins (line 476-477).

o. Details of mass spectrometry data generation and processing need to be described.

Added (line 425-451).

p. Lines 382-383; "The fold change was calculated for each prey protein as the ratio of spectral counts from replicate bait purifications to the spectral counts across all negative controls." Does this mean all negative controls for an individual protein or all negative controls for all proteins? If a protein was identified in only one replicate, was it maintained in the resulting network?

It means all negative controls for each individual protein. If a protein is identified in only one replicate, its SAINT score would be less than 0.89, so it will not be maintained in the resulting network.

q. Line 387; "...the FLAG AP-MS data set as a list of true positive interactions..." it's not clear what the authors are referring to here. Is this the "database of 77 controls and 161 samples"? If so, what evidence indicates that these are "true positive interactions"?

It is deleted in the revision.

r. Lines 392-396; To clarify, were STRING, InACT and BioGRID databases used to determine whether host proteins are known to interact with influenza virus proteins? If so, do these databases include data from the following publications? PMID: 25464832, PMID: 21715506, PMID: 26651948 (among others)

We used STING and 7 publications (PMID: 25464832, PMID: 21715506, PMID: 26651948, PMID: 26789921, PMID: 25187537, PMID: 20064372; PMID: 22810585) (line 496).

s. The authors need to describe how they performed functional enrichment analysis.

Added (line 497-500).

t. The authors need to describe the generation of stable cell lines.

Added (line 407-410).

20. Figure 3b would be better if it showed interaction between endogenous PKP2 and influenza PB1 proteins in infected cells, similar to the experiment shown in Figure 3a.

We tried experiments as suggested, however, the PB1 antibody failed to or very weakly recognized the PB1 proteins from NY/2009 and WSN/33 strains.

21. Figure Legends

a. Line 524-525; How long after infection were luciferase assays performed?

16 hr. Added in the Figure Legend (line 664).

b. Line 526; "... our laboratory database containing results from 205 selective controls..." What are 'selective' controls?

Because there are some unpublished host factors in the 205 controls, we now use 32 host factors from IAV-host interactomes, 6 empty vectors and 20 genes from our previous innate immunity protein interactome. All these gene names are listed in Supplementary Table 3. Due to the relative small size of dataset, we now adopt standard deviation to determine the outstanding hits ($\leq 2xSD$ or $\geq 2xSD$ from mean).

c. For all data, the authors should indicate how many independent replicate experiments were performed in the figure legends and whether the data in the figures shows data combined across experiments.

Added (Figs. 2c, 4a, 4b, 4e, 4f, 5a-5d, 6e, S1c, S1d, S4a, S4b, S6a, S7a-S7c, S8, S9a, S9b).

d. Figure 4d and 5c; How many cells were counted for each condition?

>80 cells for each count, which is clarified in the legends now (line 698; 720-721).

e. Figure 4f; How long after plasmid transfection were infections performed?

48 hr. Added in the Figure Legend (line 700).

f. Figure 6a-c; At what time after transfection were lysates harvested?

It is 48 hr (added in the Figure Legends).

Reviewer #3:

I was asked to comment specifically on the proteomics part of the manuscript.

The authors describe the study of the host proteins interacting with 11 proteins from one strain of the Influenza A virus, and follow up with the analysis of the interactomes of five of these proteins from two other Influenza A strains. They identify 6 novel host factors regulating viral infection and then focus their detailed analysis on plakophilin 2, demonstrating its involvement in the defense against the virus. The authors use AP-MS, which is a great methodology to determine the interactomes. The findings are interesting and biologically relevant. However,

there are several points, which should be clarified before the manuscript is accepted for publication.

We thank the reviewer's positive comments and constructive suggestions. We addressed the concerns as below.

Major concerns:

1. It is unclear from the description in the Methods section and from Figure S1a (experimental workflow) whether the analyses were performed only once, or whether there were replicates. The Methods section (page 17) states: "Two independent purifications of each tagged viral protein complex were analyzed by AP-MS" - does it mean, one analysis for the cells that were subsequently infected, and one for uninfected (Figure S1a would suggest so)? Or were there replicates of each of the infected and uninfected conditions? Replicates are obviously recommended for each tagged viral protein. Otherwise it is impossible to determine the variability between individual experiments and the results may be attributed to this variability.

There are 2 biological replicates of each tagged viral protein under each condition, the infected and uninfected, respectively. We now clarify it in the Results and Methods (line 70-71; 454-455).

2. Spectral count is probably the least reliable method of quantification. Given that the authors used cultured cells, it should have been easy to label the cells metabolically, or, if this was impossible, use another method of label-free quantification.

Spectral count has a correlation to protein abundance, but we agree with the reviewer that labelling methods, such as SILAC, would have been a more accurate methodology for quantitation. However, it is not applicable for large-scale interactome mapping in our lab. Furthermore, the widely used, well-established algorithms for analysis of protein interactome database are based on the spectral count. To address this issue, we discuss the advantage of SILAC and the limitation of AP-MS in Discussion and Methods (line 196-198; 480-484).

Minor issues:

1. Please check the spelling and grammar throughout.

Checked.

2. Page 3, line 43: I would refrain from using the word "genes" in the context of interactome, because the proteins, and not genes, are the players here.

We thank the reviewer for the suggestion. We now correct them (line 43).

3. Page 7, line 141 - a citation is missing (consistent with a previous report - which report?)
Reference added (line 143).

In summary, the reviewers' comments and suggestions have further improved our manuscript.
We believe we have addressed most concerns.

Sincerely,

Shitao Li

REVIEWERS' COMMENTS:

Reviewer #1 (Remarks to the Author):

The authors have satisfactorily addressed my comments.

I have only two minor points:

Page 7, lines 128-129; presumably this should read '... inhibited influenza viral infection without pronounced impact on cell viability'.

Page 9, line 193; Supplementary Fig. 10b should be referred to here (rather than Fig. 9b).

Reviewer #2 (Remarks to the Author):

The authors addressed all comments sufficiently.

I have no further comments on this manuscript.

Reviewer #3 (Remarks to the Author):

The authors have adequately addressed my comments, so, from my point of view, the manuscript is now suitable for publication.

Responses to review comments:

We thank all reviewers for their comments and critical reading.

Reviewer #1 (Remarks to the Author): The authors have satisfactorily addressed my comments. I have only two minor points: Page 7, lines 128-129; presumably this should read ‘... inhibited influenza viral infection without pronounced impact on cell viability’. Page 9, line 193; Supplementary Fig. 10b should be referred to here (rather than Fig. 9b).

We appreciate the reviewer for the critical reading. We change the sentence to “... inhibited influenza viral infection without pronounced impact on cell viability” (page 7, line 217-218) and Supplementary Fig. 9b to Supplementary Fig. 10b (page 9, line 288).

Reviewer #2 (Remarks to the Author): The authors addressed all comments sufficiently. I have no further comments on this manuscript.

Reviewer #3 (Remarks to the Author): The authors have adequately addressed my comments, so, from my point of view, the manuscript is now suitable for publication.